# Optimal transport unlocks end-to-end learning for single-molecule localization

**Romain Séailles**[*]
Département d'informatique, École normale supérieure (ENS-PSL), Paris, France
Université Grenoble Alpes, Inria, CNRS, Grenoble INP, LJK, Grenoble, France

**Jean-Baptiste Masson**
Decision and Bayesian Computation, Institut Pasteur, Université Paris Cité, CNRS,
Inria (Epiméthée), Paris, France

**Jean Ponce**
Département d'informatique, École normale supérieure (ENS-PSL), Paris, France
Courant Institute School of Mathematics, Computing, and Data Science,
New York University (NYU), New York, NY, USA

**Julien Mairal**
Université Grenoble Alpes, Inria, CNRS, Grenoble INP, LJK, Grenoble, France

## Abstract

Single-molecule localization microscopy (SMLM) allows reconstructing biology-relevant structures beyond the diffraction limit by detecting and localizing individual fluorophores — fluorescent molecules stained onto the observed specimen — over time to reconstruct super-resolved images. Currently, efficient SMLM requires non-overlapping emitting fluorophores, leading to long acquisition times that hinder live-cell imaging. Recent deep-learning approaches can handle denser emissions, but they rely on variants of non-maximum suppression (NMS) layers, which are unfortunately non-differentiable and may discard true positives with their local fusion strategy. In this presentation, we reformulate the SMLM training objective as a set-matching problem, deriving an optimal-transport loss that eliminates the need for NMS during inference and enables end-to-end training. Additionally, we propose an iterative neural network that integrates knowledge of the microscope's optical system inside our model. Experiments on synthetic benchmarks and real biological data show that both our new loss function and architecture surpass the state of the art at moderate and high emitter densities. Code is available at `https://github.com/RSLLES/SHOT`.

## 1 Introduction

Fluorescence microscopy remains a cornerstone tool of biological research, recording photon emissions from fluorophores (fluorescent molecules) stained onto a specimen to characterize its structure. However, light diffraction restricts the final image resolution to approximately half the wavelength of light, preventing analysis of structures or organelles feature smaller than $\sim 200\,\text{nm}$ in practice (Mc-cutchen, 1967; Schermelleh et al., 2019).

Multiple experimental techniques have been developed to surpass the diffraction limit (Hell & Wichmann, 1994; Gustafsson, 2000; Dertinger et al., 2009; Laine et al., 2023), collectively described as *super-resolution microscopy* methods. Among them, *single molecule localization microscopy* (SMLM) takes advantage of the stochastic flickering of fluorophores over a long sequence of images (Betzig et al., 1991). Compared to conventional fluorescence microscopy, the laser power is

---

[*]Corresponding author: `contact@romainseailles.fr`

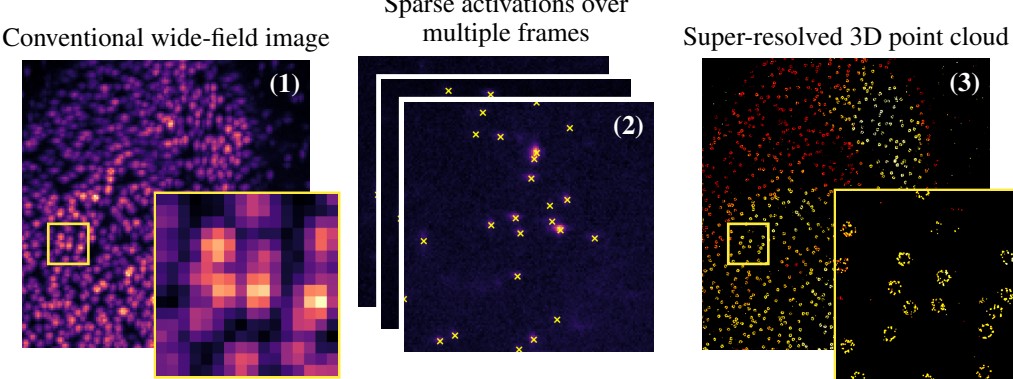

Figure 1: Illustration of the SMLM principle using our method. Data (Fei et al., 2025) show Nup96 in human bone cancer (U2-OS) cells. (1) A conventional wide-field microscope would record an image with limit resolution of $\sim 200\,\text{nm}$. (2) Instead, SMLM captures many frames where only a sparse subset of fluorophores actively emit in each one. These can be detected and localized with sub-pixel precision. (3) The union of all detections is rendered as a 3D point cloud (color encodes depth), producing a super-resolved representation of the specimen.

tuned to achieve a low density of simultaneously active fluorophores such that, with high probability, no two emitters occupy the same diffraction-limited area at the same time (Lelek et al., 2021). As modelling light propagation for point emitters in the microscope can be approximated, each emitter pixel pattern can be deconvolved into point localization with positionning error massively inferior to the diffraction limit. Accumulating detections across all frames yields a point cloud representation of the underlying specimen (Rust et al., 2006), effectively achieving super-resolution. Figure 1 illustrates this method. For additional details, on both experimental methods and deconvolution approaches see the review by Lelek et al. (2021).

However, the low-density constraint inherent to this approach limits the number of active fluorophores that can be captured in a single frame, requiring thousands of frames to reconstruct a complete specimen, which hinders live-cell imaging and the observation of dynamic processes (Heilemann et al., 2008). Consequently, high-density setups are desirable, but overlapping fluorophores within the same diffraction-limited area usually lead to uncertainties in the number of fluorophores and reduced spatial resolution, yielding deteriorated reconstruction.

Deep learning methods have shown success at handling higher densities. Top methods (Speiser et al., 2021; Fei et al., 2025) predict a detection map trained with pixel-wise objectives, and decide at inference whether a candidate exists or not by binarizing their map using a variant of non-maximum suppression (NMS) (Girshick et al., 2014). This NMS-variant uses two thresholds to (i) suppress spurious local maxima while (ii) not merging nearby emitters. We see three main issues with this framework. (1) These pixel-wise loss functions do not account for multiple emitters within the same pixel. (2) Objectives (i) and (ii) are inherently in conflict, and this issue only worsens as density increases, where the probability of multiple emitters activating simultaneously at sub-pixel distances rises. (3) The precision-recall tradeoff is difficult to tune due to the two required hand-set thresholds. Figure 6 in the Appendix illustrates problems (1) and (2).

In this paper, we frame the SMLM training objective based on one-to-one matching between predicted and true emitters, using a new loss function constructed from optimal transport (Peyré et al., 2019), and solve the decision problem at inference with a simple individual one-threshold filtering. These changes solve problem (1) by removing pixel-wise assignments in the training objective, problem (2) by removing decision pipelines based on spatial proximity like NMS, and problem (3) by using a single threshold during filtering, which directly controls the precision-recall tradeoff. Furthermore, NMS non-differentiability prevents the model from optimizing for it: discarding it allows us to benefit from the flexibility of deep neural networks at the final model layer, unlocking end-to-end learning. Additionally, inspired by the success of iterative refinement networks for optical flow estimation (Teed & Deng, 2020; Hur & Roth, 2019) we propose a novel iterative neural network architecture that

leverages a reconstruction of the expected frame given the current estimated set of fluorophores, introducing knowledge of the microscope's optic system into the model. We demonstrate that both our loss function and architecture choices improve the state of the art at both low- and high-density regimes on synthetic benchmarks and real data.

## 2 RELATED WORK

**Single-molecule localization microscopy.** SMLM has been enabled by the development of photoactivable and photoswitchable fluorophores, which allows individual molecules to emit efficiently and in a controllable manner sufficient amount of photons to be individually located (Betzig et al., 2006; Hess et al., 2006). Early tools perform detection by locating local maxima and localizing with a Gaussian estimation of the point spread function (PSF) (Patterson et al., 2010; Rust et al., 2006), assuming simplified light propagation in the microscope. Shortly after, the introduction of asymmetry along the z-axis of the PSF enabled 3D localization (Huang et al., 2008); the most common setup is to introduce astigmatism in the microscope optics, which we employ in this work. To improve localization accuracy, PSF models have transitioned from being only theory-derived to experimentally augmented, in order to incorporate effects of real light propagation in the microscope and unmodelled effect of light propagation in the cell (Babcock et al., 2012). This usually requires a pre-calibration step using specially designed fluorescent beads, which are imaged to capture how a single point of light appears at different locations. Note that while this calibration phase can be resource- and time-consuming, recent works propose live estimation of the PSF (Liu et al., 2024). 3D-DAOSTORM (Babcock et al., 2012) is a widely used classical method that uses experimentally derived PSFs, and we use it as a baseline in our comparisons.

SMLM can deliver high resolution ($10 - 20\,\text{nm}$) in optimized conditions and low phototoxicity at the cost of slow acquisition speed (Lelek et al., 2021). SIM (Gustafsson, 2000), SOFI (Dertinger et al., 2009), and eSRRF (Laine et al., 2023) trade speed for resolution, while STED (Hell & Wichmann, 1994) offers faster imaging but higher phototoxicity. MINFLUX (Balzarotti et al., 2017) provides SMLM-level resolution but lacks a large field of view and requires ultra-stable hardware (Scheiderer et al., 2025). SMLM thus remains an attractive middle ground, making high-density performance a major research focus (Lelek et al., 2021).

Deep-learning methods are widely used in fluorescence microscopy (Nehme et al., 2018; Ouyang et al., 2018; Boyd et al., 2018; Cachia et al., 2023; Li et al., 2023; Mentagui et al., 2024; Fei et al., 2025). Among these, DeepLoco (Boyd et al., 2018) uses a set formulation with a loss function based on maximum mean discrepancy (Gretton et al., 2012). DECODE (Speiser et al., 2021) combines pixel-wise detection and Gaussian-mixture localization losses, currently leading the EPFL SMLM challenge (Sage et al., 2019). More recently, LiteLoc (Fei et al., 2025) refined DECODE's architecture for improved performance. We use both DECODE and LiteLoc as baselines in our benchmarks.

**Optimal transport for set matching.** Optimal transport (Peyré et al., 2019; Villani, 2021) has become a popular tool for set matching by deep learning. Recent works in object detection (Carion et al., 2020; Zhu et al., 2021; Zhang et al., 2023; Li et al., 2022) have demonstrated success in predicting sets of variable and unknown size using bipartite matching loss functions, while other modern works have employed entropic regularization (Cuturi, 2013) to achieve fully differentiable pipelines (Zareapoor et al., 2024). By framing SMLM as a set matching problem, we draw a direct connection to this line of work — substituting objects for fluorophores — enabling the design of an end-to-end training procedure.

**Iterative refinement network.** Iterative refinement within neural networks has proven effective for tasks that benefit from sequential solution improvement (Carreira et al., 2016; Yu et al., 2023). In computer vision, Putzky & Welling (2017) have applied this approach to inverse problems such as image denoising, super-resolution, and inpainting, while Hur & Roth (2019) proposed iterative optical-flow refinement using a feedback loop with a rewarping operator. As the physics of SMLM is well understood (Etheridge et al., 2022), we show that an accurate simulator of the microscope's physics can provide similar visual feedback, enabling progressive refinement of the solution.

## 3 METHOD

### 3.1 PROBLEM FORMULATION

In this section, we first introduce the image formation model for SMLM and formulate the corresponding inverse problem as a set matching task. We then present a differentiable loss function and an iterative refinement architecture that explicitly leverages the image formation process.

**Image formation model.** An *activation* is defined as an emission event from a fluorophore within a given frame (a single fluorophore may produce several activations across multiple frames). Throughout this work, an activation is represented by a 4D vector $\boldsymbol{x} = (x, y, z, n)$, where $(x, y)$ denote the 2D coordinates in the camera frame (with the origin at the top-left corner), $z$ represents the axial coordinate relative to the focal plane, and $n$ is the photon count. Given $N$ activations within a frame, we denote the complete set as $\mathcal{X} = \{\boldsymbol{x}_i\}_{1 \leq i \leq N}$.

Diffraction within the optical system is modeled by a convolution with the *point spread function* (PSF) (Rossmann, 1969), which represents the image of a single point source. We define the PSF as a function $\mathbf{P} : \mathbb{R}^3 \longmapsto \mathbb{R}^{H \times W}$, outputting a normalized $H \times W$ image for a point source at given 3D coordinates. To ensure photon count independence, the image is normalized to sum to unity in the focal plane ($z = 0$). For a set of activations $\mathcal{X}$ in a frame, the observed image $\mathbf{H}(\mathcal{X})$ is the weighted sum of PSFs, where each weight is the photon count $n$ of the activation:

$$\mathbf{H}(\mathcal{X}) = \sum_{(x,y,z,n) \in \mathcal{X}} n\mathbf{P}(x, y, z). \tag{1}$$

The dependence of the PSF on depth $z$ enables 3D localization of activations from the observed image, see (Ovesný et al., 2014). Following Babcock & Zhuang (2017), we assume that the PSF is pre-calibrated on synthetic fluorescent beads and implemented as a collection of 3D splines. This approach is a standard tool in SMLM used in many works (Ries, 2020; Li et al., 2020; Speiser et al., 2021; Etheridge et al., 2022); see Babcock & Zhuang (2017) for further details.

**Noise model.** We adopt the noise model of Sage et al. (2019), combining shot noise (Poisson distributed), amplification noise (modeled by a Gamma distribution for EM-CCD cameras), and readout noise (normally distributed). Detailed camera parameters are in Appendix A.1. As noise is independent and identically distributed for each sensor (Fazel & Wester, 2022), it is applied independently to all pixels of $\mathbf{H}(\mathcal{X})$. We denote by $\mathbb{P}$ the distribution of images $\boldsymbol{y}$ produced by fluorophores $\mathcal{X}$ under this model such that

$$\boldsymbol{y} \sim \mathbb{P}(\mathcal{X}). \tag{2}$$

**Risk minimization formulation for set matching.** As no ground truth is available in most scientific imaging applications, supervised-learning models for SMLM have to be trained with a simulator, which is able to generate realistic $\boldsymbol{y}$ from $\mathbb{P}(\mathcal{X})$ given sets $\mathcal{X}$ of activations from a distribution $\mathcal{D}$. Our approach consists of training a neural network $f_\theta$ which directly predicts a set of activations given an observation $\boldsymbol{y}$, by minimizing the risk

$$\theta^* = \underset{\theta \in \Theta}{\arg\min} \, \mathbb{E}_{\mathcal{X} \sim \mathcal{D}, \boldsymbol{y} \sim \mathbb{P}(\mathcal{X})} \left[ \mathcal{L}(f_\theta(\boldsymbol{y}), \mathcal{X}) \right]. \tag{3}$$

Such a formulation raises two major challenges: we need to design a differentiable loss function $\mathcal{L}$ and an architecture $f_\theta$ that are appropriate to the context of SMLM.

### 3.2 OPTIMAL TRANSPORT LOSS FUNCTION

We argue that framing SMLM as a supervised-learning problem leads to a set matching formulation, for which optimal transport theory is a natural fit. To the best of our knowledge, however, this framework has not yet been applied to SMLM. Figure 2 provides an overview of our method.

Let $\mathcal{X} = \{\boldsymbol{x}_i\}_{1 \leq i \leq N}$ be the ground truth set of activations. The size of this set, $N$, is unknown and varies between frames, but it can be bounded by the physics of the fluorophore and the experimental protocol. We simulate an acquisition $\boldsymbol{y} \sim \mathbb{P}(\mathcal{X})$ and aim to retrieve $\mathcal{X}$ from $\boldsymbol{y}$.

Given $\boldsymbol{y}$, the network $f_\theta$ outputs a set of $d$ candidate activations $\hat{\mathcal{X}} = \{\hat{\boldsymbol{x}}_i\}_{1 \leq i \leq d}$ and their corresponding detection scores $\hat{\mathcal{S}} = \{\hat{s}_i \in (0, 1)\}_{1 \leq i \leq d}$. The architecture of $f_\theta$ is detailed in Section 3.3.

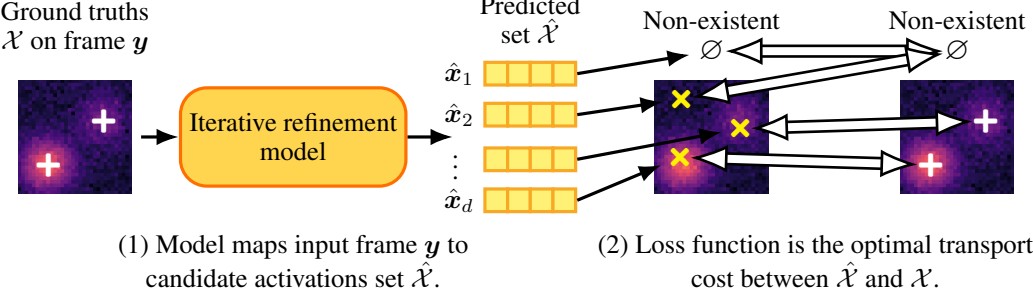

(1) Model maps input frame $\boldsymbol{y}$ to candidate activations set $\hat{\mathcal{X}}$.

(2) Loss function is the optimal transport cost between $\hat{\mathcal{X}}$ and $\mathcal{X}$.

Figure 2: Illustration of our loss function for end-to-end training. (1) Given a simulated image and its ground truth activations (see Section 3.1), our model (see Section 3.3) predicts $d$ candidate activations, each with a detection score quantifying the plausibility of its existence. (2) We solve a regularized optimal transport problem — conceptually similar to a bi-matching between ground truths and predictions — over a cost involving both localization and detection tasks. Our loss function is the optimal cost yielded by this solution.

The fixed parameter $d$ defines the maximum number of detectable activations; see Appendix A.2 for an analysis of its impact.

We first define $\boldsymbol{L}$, a squared cost matrix of size $d \times d$, whose components are:

$$\forall 1 \leq i, j \leq d, L_{i,j} = \begin{cases} (\hat{\boldsymbol{x}}_i - \boldsymbol{x}_j)^T \boldsymbol{\Sigma}^{-1} (\hat{\boldsymbol{x}}_i - \boldsymbol{x}_j) + \log \det (\boldsymbol{\Sigma}) & \text{if } j \leq N, \\ 0 & \text{otherwise,} \end{cases} \tag{4}$$

where $\boldsymbol{\Sigma} = \text{diag}(\sigma_x^2, \sigma_y^2, \sigma_z^2, \sigma_n^2)$ is a diagonal weighting matrix. Quadratic costs are a natural and principled choice for regression tasks. Extending this formulation to the negative log-likelihood of a multivariate normal distribution allows to learn $\boldsymbol{\Sigma}$ end-to-end, which can be viewed as an automatic weighting strategy that balances the difficulty of predicting each dimension, similar to the homoscedastic uncertainty weighting method proposed by Kendall et al. (2018). Experimentally, we have found $\sigma_z^2$ to be $\sim 2\times$ larger than $\sigma_x^2$ and $\sigma_y^2$ after training, which is consistent with the optical theory of confocal microscopy (Pawley, 2006).

Similarly, we define $\boldsymbol{D}$, another $d \times d$ cost matrix whose components are:

$$\forall 1 \leq i, j \leq d, D_{i,j} = \begin{cases} -\log(s_i) & \text{if } j \leq N, \\ -\log(1 - s_i) & \text{otherwise.} \end{cases} \tag{5}$$

The binary cross-entropy cost is a natural choice for detection tasks. It favors a high score $s_i$ when $\hat{\boldsymbol{x}}_i$ is paired with an element of $\mathcal{X}$ and low score otherwise, hence promoting good detection. Finally, we define the total cost matrix $\boldsymbol{C} = \boldsymbol{L} + \boldsymbol{D}$, which integrates both localization and detection tasks.

For the initial set matching problem, the optimal solution $(\hat{\mathcal{X}}^*, \hat{\mathcal{S}}^*)$ given the target $\mathcal{X}$ consists of $N$ elements identical to $\mathcal{X}$ with detection scores near 1, while the other $d - N$ elements have scores near 0. Ideally, each candidate in $\hat{\mathcal{X}}^*$ is compared to its nearest counterpart in $\mathcal{X}$ to minimize a pairwise loss. This is achieved by solving an optimal-transport problem over $\boldsymbol{C}$—effectively a bipartite matching between predictions and ground truths—where the minimal cost aggregates all pairwise contributions. Thus, our ideal loss function is the optimal-transport cost with respect to $\boldsymbol{C}$, solving:

$$\min_{\boldsymbol{\Gamma} \in \mathcal{B}} \langle \boldsymbol{\Gamma} \mid \boldsymbol{C} \rangle_{\mathcal{F}} \quad \text{where } \mathcal{B} = \left\{ \boldsymbol{\Gamma} \in \mathbb{R}_+^{d \times d} \mid \boldsymbol{\Gamma} \mathbf{1}_d = \boldsymbol{\Gamma}^\top \mathbf{1}_d = \mathbf{1}_d \right\}, \tag{6}$$

and $\langle . \mid . \rangle_{\mathcal{F}}$ is the Frobenius inner product. However, while the Hungarian algorithm can exactly solve this problem in $\mathcal{O}(d^3)$ (Kuhn, 1955), its algorithmic step is non-differentiable, which prevents end-to-end learning. We circumvent this issue by finding $\boldsymbol{\Gamma}$ through the entropy-regularized optimal transport problem, see (Cuturi, 2013), and therefore define our loss function as follows:

$$\mathcal{L}(\hat{\mathcal{X}}, \hat{\mathcal{S}}, \mathcal{X}) = \langle \boldsymbol{\Gamma}^* \mid \boldsymbol{C} \rangle_{\mathcal{F}}, \quad \text{where } \boldsymbol{\Gamma}^* = \arg\min_{\boldsymbol{\Gamma} \in \mathcal{B}} \langle \boldsymbol{\Gamma} \mid \boldsymbol{C} \rangle_{\mathcal{F}} - \epsilon H(\boldsymbol{\Gamma}), \tag{7}$$

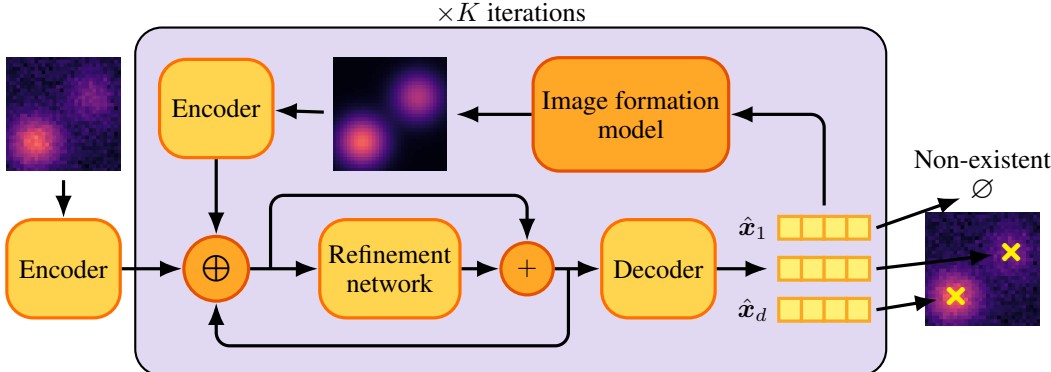

Figure 3: Illustration of our iterative refinement model. Within a classic encoder-decoder architecture, we leverage prior knowledge about the known image formation model (not learned) to simulate the expected frame given the current latent representation. This feedback is used to iteratively refine the model's inner latent representation for $K$ steps. The encoders are identical. $+$ and $\oplus$ respectively denote element-wise addition and concatenation.

$H$ is the Shannon entropy and $\epsilon$ the entropic regularization parameter. A good approximation of $\mathbf{\Gamma}^*$ in Eq. (7) can be found efficiently with a few iterations of the Sinkhorn algorithm, whose steps are differentiable with respect to the elements of $C$, enabling its use within a deep learning framework, see (Genevay et al., 2018; Mialon et al., 2021). A more detailed analysis of the impact of $\epsilon$ and a comparison with the Hungarian algorithm are available in Appendix A.4.

### 3.3 ITERATIVE REFINEMENT SCHEME

To solve Eq. (3), we investigate architectures that explicitly leverage the image formation process. To this end, we adopt an iterative architecture, an idea that has proven successful for optical flow estimation (Hur & Roth, 2019). At each iteration the network produces a set of candidate activations, turns those proposals into a simulated image, which is then used as feedback to refine the next proposals. This iterative method is illustrated in Figure 3.

Concretely, let $\boldsymbol{y}$ be the input frame of size $H \times W$. An encoder $\boldsymbol{E} : \mathbb{R}^{H \times W} \longmapsto \mathbb{R}^{C \times H \times W}$ maps $\boldsymbol{y}$ to a latent representation $\boldsymbol{z}^{(0)}$, where $C$ is a hyperparameter controlling the dimension of the latent space. A decoder $\boldsymbol{D}$ then maps latent variables to a set of candidate activations $\hat{\mathcal{X}} = \{\hat{\boldsymbol{x}}_i\}_{1 \leq i \leq d}$ and a corresponding set of detection scores $\hat{\mathcal{S}} = \{\hat{s}_i\}_{1 \leq i \leq d}$.

Given $(\hat{\mathcal{X}}, \hat{\mathcal{S}})$, we compute a reconstructed frame $\hat{\boldsymbol{y}} = \mathbb{E}[\hat{\boldsymbol{y}}|\hat{\mathcal{X}}, \hat{\mathcal{S}}]$. As the expected image from the current proposal set, $\hat{\boldsymbol{y}}$ summarizes what the model explains in the SMLM frame. Comparing $\hat{\boldsymbol{y}}$ to the original frame $\boldsymbol{y}$ supplies informative feedback, which helps the model correct errors and refine the candidates set over iterations.

Concretely, we define an iterative refinement operator $\boldsymbol{R} : \mathbb{R}^{3 \times C \times H \times W} \longmapsto \mathbb{R}^{C \times H \times W}$ which produces a residual update of the latent representation given its current estimate, the representation of the simulated frame, and the encoded original frame. Algorithm 1 shows how this proposal is updated successively over $K$ steps. During training, the final decoded output $(\hat{\mathcal{X}}^{(K)}, \hat{\mathcal{S}}^{(K)})$ is used as input for our loss function, see Section 3.2. Details about the decoder architecture and the computation of $\hat{\boldsymbol{y}}$ given $\hat{\mathcal{X}}$ and $\hat{\mathcal{S}}$ can be found in Appendix A.2.

## 4 EXPERIMENTS

**Implementation and training details.** We construct a synthetic target activations set $\mathcal{X}$ from $\mathcal{D}$ by uniformly sampling between 10 and 30 activations per frame, assigning each activation independent coordinates that are uniformly distributed across all dimensions. This guarantees that the network cannot learn any specific prior about the activation distribution.

---

**Algorithm 1** Iterative refinement architecture

---

**Require:** input frame $\boldsymbol{y} \in \mathbb{R}^{H \times W}$, encoder $\boldsymbol{E}$, decoder $\boldsymbol{D}$, refinement module $\boldsymbol{R}$, camera model $\mathbb{P}(\cdot)$, number of iterations $K \in \mathbb{N}$
**Ensure:** final proposals $(\hat{\mathcal{X}}^{(K)}, \hat{\mathcal{S}}^{(K)})$
  1: $\boldsymbol{z}^{(0)} \leftarrow \boldsymbol{E}(\boldsymbol{y})$                                         ▷ encode original frame
  2: $(\hat{\mathcal{X}}^{(0)}, \hat{\mathcal{S}}^{(0)}) \leftarrow \boldsymbol{D}(\boldsymbol{z}^{(0)})$                          ▷ decode initial proposals
  3: **for** $k = 0$ to $K - 1$ **do**
  4:     $\hat{\boldsymbol{y}}^{(k)} \leftarrow \mathbb{E}[\mathbb{P}(\hat{\mathcal{X}}^{(k)}, \hat{\mathcal{S}}^{(k)})]$         ▷ simulate reconstruction from current proposals
  5:     $\hat{\boldsymbol{z}}^{(k)} \leftarrow \boldsymbol{E}(\hat{\boldsymbol{y}}^{(k)})$                      ▷ encode reconstruction
  6:     $\boldsymbol{z}^{(k+1)} \leftarrow \boldsymbol{z}^{(k)} + \boldsymbol{R}(\boldsymbol{z}^{(k)}, \hat{\boldsymbol{z}}^{(k)}, \boldsymbol{z}^{(0)})$        ▷ refine latent iteratively
  7:     $(\hat{\mathcal{X}}^{(k+1)}, \hat{\mathcal{S}}^{(k+1)}) \leftarrow \boldsymbol{D}(\boldsymbol{z}^{(k+1)})$         ▷ decode refined proposals
  8: **end for**
  9: **return** $(\hat{\mathcal{X}}^{(K)}, \hat{\mathcal{S}}^{(K)})$

---

Following (Speiser et al., 2021), we augment $\boldsymbol{y}$ by the previous and the next frame into a tensor $\bar{\boldsymbol{y}}$ of size $3 \times H \times W$. Including these provides additional context, without introducing a too complex prior about the physics of the fluorophore, about the frame of interest, and yields improved performance. We also randomly scaled each camera parameters by a coefficient $e^{\rho}$, where $\rho \sim \mathcal{N}(0, 0.03)$: this data-augmentation "trick" increases the model robustness to experimental complexities in fully controlling and characterizing experimental parameters.

The encoder $\boldsymbol{E}$ is a two-layer U-Net (Ronneberger et al., 2015) with SiLU activations (Hendrycks & Gimpel, 2016), and LayerNorm (Ba et al., 2016) layers, with a 48-channel internal width. It maps the input to a latent image with $C = 96$ channels. The iterative refinement stage uses a similar two-layer U-Net. For the decoder $\boldsymbol{D}$, rather than the vision transformers (Dosovitskiy et al., 2021) typical of DETR-like detectors (Carion et al., 2020), we found a light CNN yields better performance (see Appendix A.2). The resulting network predicts $d = HW/4$ candidates with $\sim 3$ million parameters. As performance gains plateau after three iterations, we use $K = 2$ for all experiments.

For training, we use AdamW (Loshchilov & Hutter, 2019) for 100,000 steps with a batch size of 128 on a NVIDIA-H100 gpu, taking approximately 20h. The iterative architecture incurs a higher computational burden than single-pass models like DECODE or LiteLoc; further details about our model computational footprint for training and inference are available in Appendix A.3.

**Inference and detection-localization trade-off.** During inference, we only retain candidate activations from $\hat{\mathcal{X}}^{(K)}$ whose associated detection scores in $\hat{\mathcal{S}}^{(K)}$ exceed a user-defined threshold $\tau$ in $[0, 1]$. This simple filtering strategy makes $\tau$ an easy lever to control the precision-recall trade-off: $\tau = 0$ keeps every candidate while $\tau = 1$ discards all. By contrast, DECODE and LiteLoc use a two-threshold variant of a NMS strategy (Speiser et al., 2021; Fei et al., 2025) that may be harder to tune and harder to adapt to changing dynamics during the recording.

For the default $\tau$, we propose to maximize the $E_{3D}$ metric (Section 4) on a separate synthetic dataset from our simulator. This yields a threshold with the same detection-localization trade-off as the EPFL challenge (Sage et al., 2019). To avoid biasing Table 1, we similarly optimized DECODE's and LiteLoc's NMS parameters. For experimental data, hyperparameters can be tuned to match specific settings, ensuring consistent precision over long recordings.

**Synthetic data.** Because no ground-truth annotations exist for real SMLM acquisitions, we have performed the initial evaluation on the open synthetic datasets provided by Sage et al. (2019) on the 2016 EPFL challenge, and have adopted their set of metrics.

To evaluate candidate activations in a frame, we first solve a Hungarian assignment between ground-truths and predicted activations. A prediction is considered a *true positive* (TP) if it lies within $\pm 250\,\mathrm{nm}$ in both $x$ and $y$ directions, and $\pm 500\,\mathrm{nm}$ in z relative to its matched ground-truth (both thresholds come from the EPFL challenge). Otherwise, predictions (resp. ground truths) are labeled as *false positives* (resp. *false negatives*). Detection performance is quantified by computing *precision*, *recall* and *Jaccard Index* (*area under the curve* is not commonly employed in this field). Localization performance is evaluated by computing the *root-mean-square error* (RMSE) for TPs, for the lateral plane ($\mathrm{RMSE}_{\mathrm{lat}}$), the axial dimension ($\mathrm{RMSE}_{\mathrm{ax}}$), and all three dimensions together ($\mathrm{RMSE}_{\mathrm{vol}}$). A

Table 1: Comparative evaluation of SMLM algorithms on the EPFL 2016 challenge datasets and metrics. Densities are expressed in activations/μm/frame. For each method, means and standard deviations are estimated over four independent training seeds (3D-DAOSTORM is deterministic). *The EPFL 2016 challenge does not include a dataset with a density of 8.0; see the main text for details about its creation process.

| Density | SNR | Method | Precision ↑ | Recall ↑ | Jaccard ↑ | RMSE$_{lat}$ ↓ | RMSE$_{ax}$ ↓ | E$_{3D}$ ↑ |
|---|---|---|---|---|---|---|---|---|
| 0.2 | High | 3D-DAOSTORM | 0.964 | 0.919 | 0.914 | 11.9 | 16.9 | 0.821 |
| | | DECODE | $0.961 \pm 0.003$ | $\mathbf{0.998 \pm 0.001}$ | $0.959 \pm 0.003$ | $8.8 \pm 0.1$ | $10.7 \pm 0.1$ | $0.895 \pm 0.003$ |
| | | LiteLoc | $0.996 \pm 0.002$ | $0.987 \pm 0.001$ | $\mathbf{0.983 \pm 0.001}$ | $9.0 \pm 0.1$ | $11.7 \pm 0.1$ | $0.912 \pm 0.001$ |
| | | Ours | $\mathbf{0.998 \pm 0.002}$ | $0.978 \pm 0.016$ | $0.980 \pm 0.010$ | $\mathbf{7.5 \pm 0.4}$ | $\mathbf{10.0 \pm 3.5}$ | $\mathbf{0.920 \pm 0.007}$ |
| | Low | 3D-DAOSTORM | 0.978 | 0.835 | 0.833 | 19.3 | 29.8 | 0.685 |
| | | DECODE | $0.918 \pm 0.002$ | $\mathbf{0.978 \pm 0.001}$ | $0.903 \pm 0.002$ | $20.5 \pm 0.1$ | $26.2 \pm 0.1$ | $0.757 \pm 0.001$ |
| | | LiteLoc | $\mathbf{0.995 \pm 0.001}$ | $0.939 \pm 0.001$ | $0.934 \pm 0.001$ | $17.0 \pm 0.1$ | $25.0 \pm 0.4$ | $0.798 \pm 0.001$ |
| | | Ours | $0.985 \pm 0.001$ | $0.961 \pm 0.001$ | $\mathbf{0.947 \pm 0.001}$ | $18.8 \pm 0.2$ | $\mathbf{24.5 \pm 0.1}$ | $\mathbf{0.802 \pm 0.002}$ |
| 2.0 | High | 3D-DAOSTORM | 0.914 | 0.678 | 0.643 | 56.8 | 76.6 | 0.373 |
| | | DECODE | $0.923 \pm 0.003$ | $\mathbf{0.946 \pm 0.002}$ | $0.876 \pm 0.004$ | $32.2 \pm 0.3$ | $33.0 \pm 0.4$ | $0.706 \pm 0.004$ |
| | | LiteLoc | $\mathbf{0.993 \pm 0.001}$ | $0.863 \pm 0.002$ | $0.858 \pm 0.001$ | $30.7 \pm 0.2$ | $36.0 \pm 0.3$ | $0.699 \pm 0.001$ |
| | | Ours | $0.992 \pm 0.002$ | $0.895 \pm 0.011$ | $\mathbf{0.883 \pm 0.007}$ | $\mathbf{24.8 \pm 0.6}$ | $\mathbf{28.4 \pm 0.5}$ | $\mathbf{0.750 \pm 0.004}$ |
| | Low | 3D-DAOSTORM | 0.914 | 0.496 | 0.475 | 74.4 | 120.0 | 0.116 |
| | | DECODE | $0.859 \pm 0.034$ | $\mathbf{0.874 \pm 0.006}$ | $0.756 \pm 0.027$ | $56.4 \pm 0.3$ | $65.3 \pm 0.4$ | $0.468 \pm 0.008$ |
| | | LiteLoc | $\mathbf{0.992 \pm 0.001}$ | $0.729 \pm 0.002$ | $0.725 \pm 0.001$ | $\mathbf{46.2 \pm 0.4}$ | $63.8 \pm 0.2$ | $0.500 \pm 0.002$ |
| | | Ours | $0.973 \pm 0.003$ | $0.812 \pm 0.007$ | $\mathbf{0.794 \pm 0.005}$ | $48.4 \pm 0.6$ | $\mathbf{59.5 \pm 0.4}$ | $\mathbf{0.536 \pm 0.003}$ |
| 8.0* | High | 3D-DAOSTORM | 0.910 | 0.392 | 0.379 | 83.3 | 133.9 | 0.009 |
| | | DECODE | $0.973 \pm 0.001$ | $\mathbf{0.627 \pm 0.002}$ | $\mathbf{0.617 \pm 0.002}$ | $59.58 \pm 0.02$ | $71.5 \pm 0.5$ | $0.371 \pm 0.006$ |
| | | LiteLoc | $0.988 \pm 0.001$ | $0.557 \pm 0.003$ | $0.553 \pm 0.003$ | $60.4 \pm 0.2$ | $80.5 \pm 0.5$ | $0.319 \pm 0.004$ |
| | | Ours | $\mathbf{0.989 \pm 0.001}$ | $0.578 \pm 0.008$ | $0.574 \pm 0.008$ | $\mathbf{52.5 \pm 0.4}$ | $\mathbf{64.3 \pm 0.7}$ | $\mathbf{0.384 \pm 0.003}$ |
| | Low | 3D-DAOSTORM | 0.908 | 0.211 | 0.217 | 92.6 | 175.8 | −0.216 |
| | | DECODE | $0.93 \pm 0.03$ | $\mathbf{0.415 \pm 0.005}$ | $\mathbf{0.402 \pm 0.009}$ | $80.25 \pm 0.04$ | $105.2 \pm 1.0$ | $0.090 \pm 0.007$ |
| | | LiteLoc | $0.986 \pm 0.001$ | $0.339 \pm 0.002$ | $0.338 \pm 0.002$ | $76.1 \pm 0.1$ | $110.1 \pm 0.8$ | $0.055 \pm 0.002$ |
| | | Ours | $\mathbf{0.983 \pm 0.002}$ | $0.376 \pm 0.008$ | $0.374 \pm 0.008$ | $\mathbf{74.3 \pm 0.5}$ | $\mathbf{99.4 \pm 1.1}$ | $\mathbf{0.103 \pm 0.002}$ |

global performance metric called *3D efficiency* (E$_{3D}$) is then defined as:

$$E_{3D} = \frac{E_{ax} + E_{lat}}{2} \quad \text{where} \quad \begin{cases} E_{lat} = & 1 - \sqrt{(1 - \text{Jaccard})^2 + \alpha_{lat}^2 \text{RMSE}_{lat}^2}, \\ E_{ax} = & 1 - \sqrt{(1 - \text{Jaccard})^2 + \alpha_{ax}^2 \text{RMSE}_{ax}^2}, \end{cases} \quad (8)$$

$\alpha_{lat} = 1.0 \, \text{nm}^{-1}$ and $\alpha_{ax} = 0.5 \, \text{nm}^{-1}$, following definitions of the EPFL challenge. All metrics are computed frame by frame and averaged.

Our benchmark includes 3D-DAOSTORM (Babcock et al., 2012), DECODE (Speiser et al., 2021) and LiteLoc (Fei et al., 2025). All algorithms are evaluated on the open-access EPFL 2016 challenge datasets (Sage et al., 2019), all with astigmatism PSFs. To assess performance in a very high-density regime, we have synthesized a density-8.0 benchmark by temporally binning groups of 4 frames in the original density-2.0 sequences. For each newly binned frame, we have re-sampled camera noise using the known camera parameters. This extra step prevents an artificial signal-to-noise ratio (SNR) improvement caused by the frame-averaging process.

Results are reported in Table 1. We observe that while our approach yields lower recall than the other methods, it preserves excellent precision and almost always achieves the lowest RMSE in all spatial dimensions. Most notably, it also outperforms all competitors on the E$_{3D}$ metric for all densities and SNRs, establishing itself as the most balanced method with respect to this criterion.

**Real data.** We have evaluated our method on three publicly available datasets, all of which provide beads for calibrating their astigmatic PSFs. The Tubulin and NPC-Nup107 datasets from Li et al. (2018) depict, respectively, the microtubule network and nuclear pore complexes in U2OS cells. The NPC-Nup96 dataset from Fei et al. (2025) also features nuclear pore complexes in the same cell line. All datasets were acquired with conventional SMLM activation densities; therefore, to test our method's robustness at higher densities, we applied 16-frame temporal binning to Tubulin and NPC-Nup107 and 32-frame binning to NPC-Nup96. We refer to the temporally-binned versions as T16-Tubulin, T16-NPC-Nup107, and T32-NPC-Nup96. Note that this approach is an imperfect proxy for truly high-density imaging, as it improves the SNR via noise averaging.

Figure 4 compares 3D SMLM reconstructions, rendered with SMAP (Ries, 2020), for 3D-DAOSTORM (Babcock et al., 2012), LiteLoc (Fei et al., 2025) and our method. We chose to

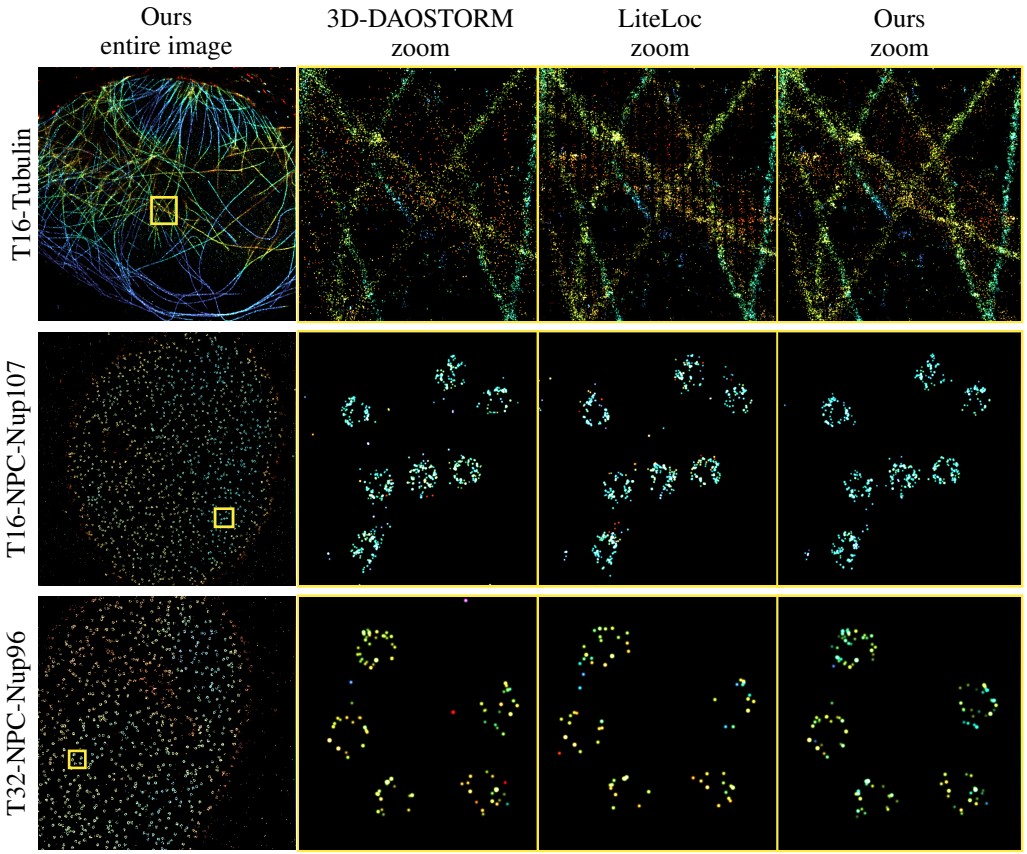

Figure 4: Qualitative comparison of SMLM methods on real data. Although ground truths are unavailable, results show that our approach yields fewer grid-reconstruction artifacts (line 1), improved depth estimation consistency (line 2), and more accurate nuclear pore complex reconstruction (line 3). Refer to the main text for a more thorough discussion.

include LiteLoc over DECODE because the former delivers comparable or slightly better performance. In addition, the authors of DECODE note that their method may benefit from an extra filtering step applied to the predicted uncertainties associated with each activation. However, this post-processing step requires selecting additional arbitrary thresholds that are difficult to tune, making it challenging to perform a fair and objective comparison with other methods.

On the T16-Tubulin dataset, our algorithm yields a higher-fidelity reconstruction than 3D-DAOSTORM and eliminates the artifacts that appear with LiteLoc. On the T16-NPC-Nup107 dataset, all methods recover comparable structures; however, our method delivers more consistent depth estimates (as indicated by colors), whereas 3D-DAOSTORM and LiteLoc exhibit spatially varying detections. On the T32-NPC-Nup96 dataset, our approach reconstructed NPC's structures with clear greater fidelity than LiteLoc and 3D-DAOSTORM.

Quantitatively, the absence of ground-truth data prevents the use of the metrics introduced in Section 4. To evaluate the resolution and fidelity of a reconstructed super-resolution image, we adopted two widely used metrics: *Fourier ring correlation* (FRC) (Banterle et al., 2013) and the *resolution-scaled Pearson's coefficient* (RSP) Culley et al. (2018). FRC reconstructs two super-resolution images by splitting localizations into two subsets, computing their Fourier transforms, and then measuring the correlation of their spatial frequency signals against each other. The resulting curve provides an estimate of the spatial frequency at which signal can no longer be distinguished from noise (Banterle et al., 2013). RSP is defined as the Pearson correlation coefficient between the reconstructed super-resolution image and a reference image, typically the mean of all raw wide-field frames. Values close to one indicate strong agreement between the reconstruction and the reference.

Table 2: Quantitative results on real datasets. We temporally binned them to simulate very high-density setups. Our method consistently scored first in those denser regimes.

| Dataset | Bin size | Method | FRC (nm) ↓ | RSP ↑ |
|---|---|---|---|---|
| Tubulin (Li et al., 2018) | ×1 | LiteLoc | **29.7 ± 0.3** | **0.708** |
| | | Ours | 31.9 ± 0.2 | 0.692 |
| | ×16 | LiteLoc | 63.0 ± 0.8 | 0.649 |
| | | Ours | **58.1 ± 1.1** | **0.672** |
| NPC-Nup107 (Li et al., 2018) | ×1 | LiteLoc | 19.3 ± 0.3 | **0.696** |
| | | Ours | **18.8 ± 0.4** | 0.686 |
| | ×16 | LiteLoc | 25.9 ± 0.3 | 0.682 |
| | | Ours | **22.1 ± 0.1** | **0.684** |
| NPC-Nup96 (Fei et al., 2025) | ×1 | LiteLoc | **29.8 ± 0.1** | **0.713** |
| | | Ours | 31.7 ± 0.1 | 0.693 |
| | ×32 | LiteLoc | 71.5 ± 0.5 | 0.671 |
| | | Ours | **44.2 ± 0.4** | **0.689** |

Table 3: Ablation study of our different modules over EPFL synthetic with high SNR and a density of 2.0. ✖ means we used DECODE's original loss function or model architecture.

| Iterative arch. | OT loss func. | Jaccard ↑ | $RMSE_{vol}$ ↓ | $E_{3D}$ ↑ |
|---|---|---|---|---|
| ✖ | ✖ | 0.876 ± 0.004 | 47.9 ± 0.5 | 0.705 ± 0.004 |
| ✖ | ✔ | 0.867 ± 0.004 | 39.6 ± 0.3 | 0.740 ± 0.002 |
| ✔ | ✖ | 0.854 ± 0.005 | 45.4 ± 0.7 | 0.703 ± 0.005 |
| ✔ | ✔ | **0.883 ± 0.007** | **39.2 ± 0.5** | **0.750 ± 0.004** |

Results with these metrics on real datasets are reported in Table 2. In dense-activations regimes, our approach consistently yields lower FRC and higher RSP values than other methods, confirming the visual improvements illustrated in Figure 4.

**Ablation study.** We have conducted an ablation study on synthetic data to validate the effectiveness of our loss function and our iterative architecture. Results are reported in Table 3. It can be seen that the loss function drives most of the improvement, with our iterative architecture providing a modest boost. Given the additional memory and compute overhead of our architecture, a lightweight variant that retains only the optimal loss function can be considered for deployment scenarios with constrained resources.

## 5   DISCUSSION AND CONCLUDING REMARKS

We have presented a novel deep-learning SMLM method that surpasses existing methods in medium and high-density regimes, all without the need for handcrafted layers. By enabling faster data acquisition, our approach extends SMLM's temporal resolution, allowing more accurate observation of rapid biological processes but also stable inference precision to degrading conditions induced by evolution of recording parameters during time. Furthermore, the integration of optimal transport theory to SMLM could open a path to new localization algorithms.

The main limitation of our method is the longer training and inference times that result from its iterative design. However, training is a one-time cost per experimental setup, and inference remains fast enough ($\sim 200\,\mathrm{fps}$ on a modern GPU) to let biologists run multiple experiments sequentially with minimal delay. Another systemic limitation shared by most top-performing methods is the dependence for precise PSF calibration (Lelek et al., 2021). Future work could focus on robust methods invariant to PSF variations, pursue blind SMLM super-resolution without sacrificing precision or include PSF optimization to the microscopy setup design during training and to adapt to cellular based optical anomalies affecting the PSF at inference time.

## ACKNOWLEDGEMENTS

We thank Luca Calatroni for fruitful discussions and valuable feedback. This work was supported by the French government through the "France 2030" program, in particular by the PR[AI]RIE-PSAI project (ref. ANR-23-IACL-0008). Julien Mairal was also supported by an ERC grant (ref. 101087696, APHELEIA project), and Jean Ponce was supported in part by the Louis Vuitton/ENS chair in artificial intelligence, the Institute of Information & Communications Technology Planning & Evaluation (IITP) grant funded by the Korean Government (MSIT) (ref. RS-2024-00457882, National AI Research Lab Project), and a Global Distinguished Professorship at the Courant Institute of Mathematical Sciences and the Center for Data Science at New York University. We received access to the computational resources of IDRIS from GENCI (ref. AD010616282).

## ETHICS STATEMENT

In this work, we explore new model architecture and loss function to improve single-molecule localization microscopy (SMLM) reconstruction. We do not anticipate ethical or societal harms: the work is computational only and all biological data used are public and were used according to their licenses. We believe that by improving SMLM reconstruction and releasing our code openly, this work can broadly benefit biological research and make advanced tools accessible to communities worldwide.

## REPRODUCIBILITY STATEMENT

The project repository (`https://github.com/RSLLES/SHOT`) includes all requirements to reproduce our results. We provide the full source code and model implementation, all datasets are publicly available, training and evaluation scripts are provided with all hyperparameters set (including random seeds), and python environment specification are supplied, making all experiments from Section 4 reproducible.

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

# A APPENDIX

## A.1 IMAGE FORMATION PROCESS

SMLM experimental setups typically employ either Electron-Multiplying CCD (EM-CCD) or scientific CMOS (sCMOS) cameras. Their sensor converts incident photons into a digital intensity value (ADU) through a sequence of physical processes, each of which introduces noise.

Let $n$ be the incident photon count on the camera sensor. Initially, photon detection is modeled as a Poisson process — known as *shot noise* — with a mean proportional to $n$ and the quantum efficiency (QE), and an offset known as the spurious charge ($c$):

$$n_{e,1} \sim \mathcal{P}\left(\text{QE} \times n + c\right). \tag{9}$$

EM-CCD cameras introduce an additional amplification stage, modeled as a Gamma distribution with parameters $n_{e,1}$ and the electromagnetic gain (EM):

$$n_{e,2} \sim \Gamma(n_{e,1}, \text{EM}) \text{ for EM-CCD camera, or } n_{e,2} = n_{e,1} \text{ for sCMOS camera.} \tag{10}$$

Subsequently, *read noise* is modeled by a normal distribution with mean $n_{e,2}$ and standard deviation $\sigma_R$:

$$n_{e,3} \sim \mathcal{N}(n_{e,2}, \sigma_R). \tag{11}$$

Finally, the analog-to-digital conversion process yields the observed ADU, scaled by the electrons per ADU ($e_{\text{ADU}}$) and offset by a baseline ($B$):

$$y = \min\left(\left\lfloor \frac{n_{e,3}}{e_{\text{ADU}}} \right\rfloor + B \, , \, 65535\right) \tag{12}$$

No algebraic solution exists for the resulting distribution relating $n$ and $y$ (Ryan et al., 2021). Table 4 presents the parameters for two commonly used cameras in SMLM: the *Evolve Delta 512* camera for the Tubulin and NPC-Nup107 datasets (Li et al., 2018) and the *Dhyana 400BSI V3* camera for the NPC-Nup96 dataset (Fei et al., 2025).

## A.2 ARCHITECTURE DETAILS

**Decoder architecture.** The decoder maps a latent representation $z$ of the image - implemented as a $C \times H \times W$ tensor - to a set of $d$ activations, implemented as a $d \times 5$ matrix (one activation contains five elements: the three spatial coordinates $(x, y, z)$, the number of emitted photons $n$ and the detection score $s$).

| Range factor | Jaccard ↑ | RMSE$_{vol}$ ↓ | E$_{3D}$ ↑ |
|:---:|:---:|:---:|:---:|
| 1.0 | 0.869 | 47.0 | 0.710 |
| 1.1 | 0.874 | 45.1 | 0.722 |
| 1.2 | 0.880 | 42.6 | 0.737 |
| 1.5 | **0.889** | **39.8** | **0.750** |
| 2.0 | 0.884 | 40.1 | 0.749 |
| 3.0 | 0.880 | 41.3 | 0.738 |

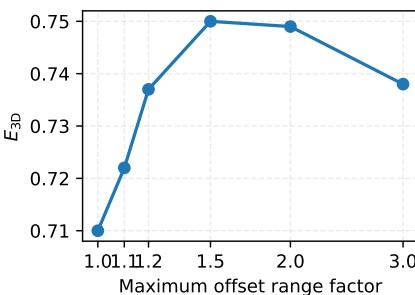

Figure 5: Evaluation of the impact of the maximum offset range of our CNN decoder. The maximum prediction range is controlled by a range factor; the final range equals the range factor multiplied by the output pixel size (which is $2\times$ larger than the original image pixel size). Our results show that a range factor of 1.5 times the output pixel size, i.e. $3\times$ the original pixel size, yields the best performance. Reported results are computed on the MT0N1HDAS dataset.

As we aim to predict a set from an image, and given the recent success of object detection by transformer architectures (Carion et al., 2020), we have considered using a vision transformer (ViT) (Dosovitskiy et al., 2021). However, this architecture produced unsatisfactory results, see Table 5. We attribute this to two factors:

1. SMLM requires sub-pixel precision, but each ViT's token spans the whole image, so a small prediction error can severely affect the output.

2. The individual localization-detection problem is highly local. Hence, ViT's global attention mechanism offers little benefit.

Therefore, we propose a convolution-based decoder architecture, composed of $2 \times 2$ max-pooling layers and residual blocks (He et al., 2016). We experimented with different numbers of max-pooling operations and found that a single max-pooling layer, followed by a residual block and an element-wise output layer yields the best results, see Table 5.

Formally, given the latent image $\boldsymbol{z}$, our decoder is defined as $\boldsymbol{D} : \mathbb{R}^{C\times H\times W} \longmapsto \mathbb{R}^{5\times H/2\times W/2}$, mapping a latent variable to a $H/2 \times W/2$ map with 5 channels, where each pixel is an activation prototype. Consider a single pixel $i$ of $\boldsymbol{D}$'s output, and let $(\tilde{x}_i, \tilde{y}_i)$ be its 2D coordinates in the camera coordinate system. The five elements output for this pixel encode the characteristics of the underlying candidate activation: the detection score $\hat{s}_i$, the relative lateral coordinates $(\Delta \hat{x}_i, \Delta \hat{y}_i)$, the depth $\hat{z}_i$ and the number of emitted photons $\hat{n}_i$. The absolute lateral coordinates $(\hat{x}_i, \hat{y}_i)$ are reconstructed by summing $(\Delta \hat{x}_i, \Delta \hat{y}_i)$ with $(\tilde{x}_i, \tilde{y}_i)$. The magnitude of the relative coordinate offsets $(\Delta \hat{x}, \Delta \hat{y})$ predicted by the decoder is set to 1.5 times the output pixel size (or in other words $3\times$ the pixel size of the original image, given the final max pooling). This extended range permits multiple activations to be mapped within a single pixel area, as neighbouring activations can contribute to their surroundings. Figure 5 shows experiments for various magnitude of the coordinate offsets. With a factor of $1\times$ the output pixel size, each output pixel can predict locations only on the exact surface it covers. In this regime, the optimal transport solution reduces to an identity pixel-wise mapping.

Table 4: Reported parameters of two typical cameras used in SMLM by their manufacturer.

| Parameter | Evolve Delta 512 | Dhyana 400BSI V3 |
|:---|:---:|:---:|
| Camera type | EMCCD | sCMOS |
| Quantum efficiency (QE) | 0.90 | 0.95 |
| Spurious charge ($c$) | 0.002 | 0.002 |
| EM gain (EM) | 300 | — |
| Readout noise ($\sigma_R$) | 74.4 | 1.535 |
| Electrons per ADU ($e_{ADU}$) | 45 | 0.7471 |
| ADU baseline (B) | 100 | 100 |

Table 5: Evaluation of different decoder architectures with different number of predicted candidates over the MT0N1HDAS dataset (Sage et al., 2019), with standard deviations computed for three different training seeds. CNNs differ by the architecture of their head module, composed of alternating $2 \times 2$ max-pooling layers and double-convolution blocks.

| Architecture | $d$ | Parameters↓ | Jaccard ↑ | RMSE$_{\text{vol}}$ ↓ | E$_{\text{3D}}$ ↑ |
|---|---|---|---|---|---|
| CNN | $HW$ | **2.31M** | $0.866 \pm 0.003$ | $41.4 \pm 0.7$ | $0.732 \pm 0.002$ |
| | $HW/4$ | 2.81M | $\mathbf{0.883 \pm 0.007}$ | $\mathbf{39.2 \pm 0.5}$ | $\mathbf{0.750 \pm 0.004}$ |
| | $HW/16$ | 3.48M | $0.875 \pm 0.008$ | $40.5 \pm 0.5$ | $0.739 \pm 0.002$ |
| ViT | $HW/4$ | 5.95M | $0.852 \pm 0.006$ | $40.6 \pm 0.7$ | $0.722 \pm 0.001$ |

Table 6: Multiply-Accumulate operations and number of parameters of our model subparts.

| Subpart | Multiply-Accumulate operations | Parameters |
|---|---|---|
| Encoder | 1.03 GMac | 1.05 M |
| Decoder | 512.56 MMac | 499.4 k |
| Residual Network | 1.87 GMac | 1.26 M |
| Renderer | 94.64 MMac | 0 |

Finally, the output is formatted into a candidate set $\hat{\mathcal{X}} = \{(\hat{x}_i, \hat{y}_i, \hat{z}_i, \hat{n}_i)\}_{1 \leq i \leq d}$ and a detection scores set $\mathcal{S} = \{\hat{s}_i\}_{1 \leq i \leq d}$. We integrate this reconstruction process into the decoder, meaning $\boldsymbol{D}(z) = (\hat{\mathcal{X}}, \hat{\mathcal{S}})$.

**Differentiable simulation within our model.** During inference, our algorithm selects a subset of candidate detections by thresholding their confidence scores. However, this operation is non-differentiable, preventing direct gradient propagation during training. To mimic this behaviour while retaining differentiability, we replace it by a soft weighting that scales the photon count of each candidate by its detection confidence. For each candidate $\boldsymbol{x}_i$ in $\hat{\mathcal{X}}$, the network outputs the 3D coordinates $(\hat{x}_i, \hat{y}_i, \hat{z}_i)$, the raw photon count $\hat{n}_i$, and a detection confidence $\hat{s}_i \in (0, 1)$. We choose to modulate the photon count by the confidence, producing the weighted activation

$$\tilde{\boldsymbol{x}}_i = \left(\hat{x}_i, \; \hat{y}_i, \; \hat{z}_i, \; \hat{s}_i \, \hat{n}_i\right),$$

and the set of all such activations is denoted $\tilde{\mathcal{X}} = \{\tilde{\boldsymbol{x}}_i\}_{i=1}^{d}$. This causes activations with low detection scores to have a number of emitted photons near zero, making them almost non-existent, while keeping almost untouched activations with a detection score close to one, mimicking the effect of a hard threshold while remaining fully differentiable.

After derivation, the expected image $\hat{\boldsymbol{y}}$ is obtained by:

$$\hat{\boldsymbol{y}} = \mathbb{E}[\hat{\boldsymbol{y}}|\tilde{\mathcal{X}}] = \frac{\text{QE} \times \text{EM}}{e_{\text{ADU}}}\mathbf{H}(\tilde{\mathcal{X}}) + B, \tag{13}$$

and with EM $= 1$ for sCMOS camera. $\tilde{\boldsymbol{y}}$ is an end-to-end differentiable approximation of the reconstructed output, and can be used inside our iterative refinement scheme.

### A.3 COMPUTATIONAL FOOTPRINT

Training is performed using an NVIDIA H100 GPU using the AdamW optimizer with a learning rate of $4 \times 10^{-4}$, a weight decay of $0.01$, and a cosine annealing scheduler. We chose a batch size of 128 to maximize GPU usage, filling all $80\,\text{GB}$ of VRAM. It can be lowered using smaller batch sizes or gradient accumulation.

We trained for 14 hours 100 epochs of 1024 steps each, totaling approximately 100,000 steps. Excellent results (E$_{\text{3D}} \geq 0.72$ on EPFL's density=2.0 and high SNR dataset) are achieved after only 20 minutes of training, at around 2000 steps.

During inference, a batch size of 16 produces a peak VRAM usage of $8.7\,\text{GB}$ and processes 2500 64x64 images in $30\,\text{s}$, or $12\,\text{ms/frame}$.

Table 7: Evaluation of different algorithms for solving the optimal transport problem used during the computation of our loss function. Results are reported for the MT0N1HDAS dataset (Sage et al., 2019), with standard deviations computed for three different training seeds.

| Algorithm | | Jaccard ↑ | RMSE$_{vol}$ ↓ | E$_{3D}$ ↑ |
|---|---|---|---|---|
| | $\epsilon = 10^{-2}$ | $0.531 \pm 0.007$ | $67.0 \pm 5.2$ | $0.397 \pm 0.011$ |
| | $\epsilon = 10^{-3}$ | $0.883 \pm 0.006$ | $42.0 \pm 1.3$ | $0.739 \pm 0.002$ |
| Sinkhorn's | $\epsilon = 10^{-4}$ | $0.883 \pm 0.007$ | $39.2 \pm 0.5$ | $\mathbf{0.750 \pm 0.004}$ |
| | $\epsilon = 10^{-5}$ | $0.877 \pm 0.012$ | $39.1 \pm 1.1$ | $0.747 \pm 0.002$ |
| Hungarian | | $0.873 \pm 0.007$ | $39.7 \pm 0.9$ | $0.742 \pm 0.001$ |

Table 8: Evaluation of the robustness of our model to domain mismatch. Multiplicative noise of increasing strength, sampled from a zero-mean log-normal distribution, is applied to the simulator's camera parameters. Our model demonstrates strong resilience to this perturbation.

| Jitter strength | Jaccard ↑ | RMSE$_{vol}$ ↓ | E$_{3D}$ ↑ |
|---|---|---|---|
| $\sigma = 0$ | $0.956 \pm 0.001$ | $34.31 \pm 0.26$ | $0.811 \pm 0.001$ |
| $\sigma = 0.03$ | $0.958 \pm 0.002$ | $34.52 \pm 0.20$ | $0.810 \pm 0.001$ |
| $\sigma = 0.10$ | $0.957 \pm 0.001$ | $34.50 \pm 0.25$ | $0.810 \pm 0.001$ |
| $\sigma = 0.30$ | $0.954 \pm 0.001$ | $35.21 \pm 0.32$ | $0.805 \pm 0.002$ |

Table 6 shows an overview of the computational resources for each subpart of our model.

### A.4 REGULARIZED OPTIMAL TRANSPORT

As explained in the main text, we solve the regularized optimal transport problem of Eq. (7) with Sinkhorn's algorithm. Our motivations are both analytical and computational: compared to the standard bipartite matching, Sinkhorn's algorithm avoids the need for stop-gradient operations, is differentiable, and is computationally efficient on GPUs. In our implementation, we run Sinkhorn's algorithm in log space and compute gradients automatically via PyTorch's autograd module. Our implementation uses 20 iterations, as we have found that additional iterations do not improve performance. Additionally, we have included a masking step to ensure that candidates are only assign to target activations if they are capable of reaching it within their limited prediction range. The observed performance boosts for increased range factors highlight the benefits of using optimal transport rather than pixel-wise assignments.

Table 7 compares results for various regularization constant $\epsilon$ in regularized optimal transport problem. We also report results with a bipartite matching performed by the Hungarian algorithm, that yields lower performance. We observe that smaller values for $\epsilon$ yield better performance, with no improvement beyond $\epsilon = 10^{-4}$; thus our we set $\epsilon = 10^{-4}$ in our implementation. Note that our implementation of the Sinkhorn's algorithm includes the common practical heuristic of scaling $\epsilon$ by the median of the cost matrix (Flamary et al., 2021).

### A.5 ROBUSTNESS TO CAMERA PARAMETERS MISMATCH

We have conducted a study to analyze the robustness of our model to mismatch with respect to all camera parameters listed in Table 4.

To this end, we have generated a synthetic dataset of 2048 frames with a mean density of 2.0, each rendered with camera parameters independently jittered by noise from a log-normal distribution, i.e. scaled by $e^\rho$ where $\rho \sim \mathcal{N}(0, \sigma)$. Note that we apply a similar data augmentation process during training, with $\sigma = 0.03$.

We have evaluated our model performance under increasing noise strength, to assess performance for increasing domain gap between training and test data. The results are reported in Table 8. Interestingly, our model appears extremely resilient to this type of mismatch. From an architecture standpoint, the use of LayerNorm and 2D convolutions without additive bias makes the network insensitive to

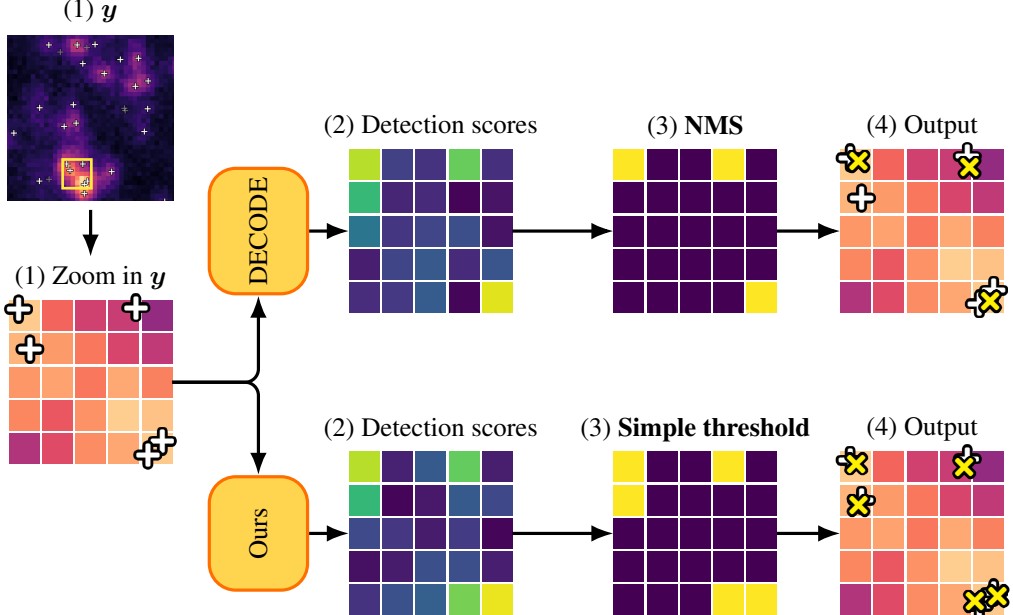

Figure 6: Toy illustration of the negative effects of pixel-wise loss functions and NMS post processing. (1) A synthetic image $y$ with a zoomed version. White crosses represent the ground truth emitters. (2) DECODE's pixel-wise loss function enforces a high score for the bottom-right pixel, but does not consider that it is shared by two targets. By contrast, our optimal transport loss function also assigns a high score to a neighboring pixel, which can contribute to its neighborhood with an extended localization range. (3) DECODE's NMS variant merges the two adjacent predictions in the top-left corner if one of the prediction scores isn't high enough to be automatically retained. By contrast, our simple single-threshold filtering keeps all scores above the threshold, regardless of their spatial distribution. (4) Models' final outputs, showing the predicted emitters for each method with yellow crosses.

scaling. We hypothesize that this architectural choice paired with the small data augmentation during training results in remarkably stable performance with respect to this issue.

