# OpenReview forum: "Optimal transport unlocks end-to-end learning for single-molecule localization"
_ICLR.cc/2026/Conference — ICLR 2026 Poster_

### Official Review · Reviewer_pzpv · 2025-10-25

**Soundness:** 2
**Presentation:** 2
**Contribution:** 2
**Rating:** 2
**Confidence:** 5

**Summary:**

This paper proposes a novel method for single-molecule localization microscopy (SMLM). The authors formulate the SMLM task as a set matching problem and employ the optimal transport cost as the loss function.
Furthermore, the proposed method utilizes a refinement model that integrates embedded features extracted from the input image, the previous prediction step, and a reconstructed image generated through an image formation model. This design effectively incorporates prior knowledge derived from an approximate point spread function into the molecular localization process.

Experimental results demonstrate that the proposed framework outperforms existing deep learning–based approaches quantitatively and qualitatively, particularly under high-density conditions.

**Strengths:**

The paper is clearly written and easy to follow, with a well-articulated background on single-molecule localization microscopy (SMLM). The formulation of SMLM as a set matching problem and the use of the optimal transport (OT) distance as a loss function are original and technically interesting. The refinement model is also a valuable contribution: it effectively integrates features from the input image, the previous prediction step, and a reconstructed image generated from the previous output, while maintaining a simple and lightweight network architecture.
Furthermore, the ablation study demonstrates the effectiveness of both the loss formulation and the proposed modeling approach.

**Weaknesses:**

While the use of the optimal transport (OT) distance for SMLM is original, its motivation is not clearly articulated. The paper refers to the proposed framework as “end-to-end learning,” but the meaning of this term is somewhat ambiguous in this context (see comments in Section 7-1 for further detail). In addition, the paper criticizes LiteLoc in the Introduction for using an NMS layer, but LiteLoc does not actually employ such a layer. As a result, it is unclear what specific limitation or issue of LiteLoc the proposed method aims to address.

In addition, the paper presents two seemingly independent contributions—a new loss function and a new network architecture—but the connection between them is not clearly established. The architecture does not appear to leverage the properties of the proposed loss, which weakens the methodological coherence of the work. The authors should better justify why both components are included in a single framework and discuss whether one could be used meaningfully without the other.

Furthermore, the training process is insufficiently described. In particular, it remains unclear what the network (especially the decoder) outputs and what training data correspond to these outputs (see also the suggestion in Section 7-2). This lack of clarity makes it difficult to assess the effectiveness of using the OT distance as a loss function.

**Questions:**

7-1. What is the intended meaning and benefit of “end-to-end learning” in this paper?

It is true that, in general, loss functions for set-matching problems are non-smooth, making it difficult to compute gradients through cost functions that depend on the network outputs, such as coordinates or confidence scores.
However, although the authors describe SMLM as a set-matching problem, the estimation in this work appears to be performed in a pixel-wise manner, where each predicted pixel can be directly compared with its corresponding ground-truth pixel. In this case, solving a set-matching problem is unnecessary, and the network can be trained end-to-end as in DECODE.

Conversely, using the OT distance as a loss function may actually hinder end-to-end learning rather than enable it. It is therefore unclear why a set-based formulation is necessary or what specific benefits it is expected to provide.

In addition, the idea of formulating SMLM as a set-matching problem is closely related to [1], where the maximum mean discrepancy (MMD) is used as a loss function to quantify the difference between the estimated and ground-truth molecule sets, while the estimation is performed for the whole image rather than in a pixel-wise manner.

[1] Boyd, N.; Jonas, E.; Babcock, H.; Recht, B. DeepLoco: Fast 3D Localization Microscopy Using Neural Networks. 2018. https://doi.org/10.1101/267096.


Suggestion
7-2. Clarify the network output and loss design.
Although the authors state that the number of candidates d is a hyperparameter, it appears to depend on the input image size (H, W), with predictions made for every four pixels. From the appendix, each pixel’s offset is up to three times larger than a pixel, which markedly differs from prior works. Since the estimation is performed pixel-wise, the ground-truth coordinate set can also be converted into pixel-wise labels—indicating whether a molecule exists and, if so, its offset within the pixel. In this case, the prediction and ground-truth representations share the same structure and ordering, and the OT loss effectively reduces to a sum of pixel-wise losses when the OT map becomes the identity. The authors should clarify how this formulation differs from a conventional pixel-wise loss, possibly by analyzing the learned transport map.

7-3. Analyze OT map
The transport cost design suggests that information from neighboring pixels might also contribute to molecular localization. Examining the OT map could help explain why, even with lower Jaccard scores, the estimated coordinates remain close in high-density cases (as in Table 1, 8.0). An ablation study comparing the OT loss with a simple pixel-wise loss would clarify the true contribution of OT and the extended offset range.

Minor Comments

- On page 7, line 366, the citation of DECODE refers to a different paper. The original DECODE paper is by Speiser et al., Nature Methods, 2021.
- Plese define RMSE_vol in Table 3.
- The reported E_{3D} values for DECODE differ by 0.001 between Table 1 and Table 3. Please clarify the reason for this discrepancy.

---

> ### Author Response · Authors · 2025-11-21
> **Response part 1/3**
>
> **Optimal transport motivation.**
> In typical object detection tasks on natural images, object size extend over multiple pixels, and different objects are often separated by large pixel gaps. By contrast, SMLM requires individual sub-pixel localisation of a large number of emitters whose centers can be less than one-pixel.
> To quantify this, we have analyzed the MT0N1HDAS dataset: 15\% of activations have a neighbor within a 100nm $\times$ 100nm pixel window, i.e. are close enough to belong to the same pixel. In our 8‑density dataset, this proportion rises to 30\%.
> In this context, the pixel‑wise formulation from DECODE potentially runs into problems: which offset to predict when a pixel is responsible for predicting several emitters? How should the detection‑score map reflect the number of emitters? We are not claiming that this difficulties cannot be solved, but we believe that adding the corresponding mechanisms introduces additional complexity on top of this formulation.
>
> The SMLM task is a set prediction problem. If it is possible to predict directly this set (which is what we do), we believe that it is a reasonable principle to do it directly, rather than considering two independent stages. Note that this line of work has also been explored in modern object detection pipelines with transformers, see our Related Work section / see (Carion
> et al., 2020; Zhu et al., 2021; Zhang et al., 2023; Li et al., 2022).
>
> We thank the reviewer for pointing out the lack of clarity of our presentation on this issue, and we have added clarifying comments at line 75 in the main text.
> Part of the confusion arise from the design of our decoder, that uses a CNN architecture instead of a transformer architecture like most modern object detection models that rely on optimal transport-based loss functions.
> We invite the reviewer to read our answer to Reviewer's 4CME about our motivation for this choice of a decoder.
> We have also added extra comments on the design of our decoder in Appendix 2 to clarify how we ended up with this architecture.
>
> **Evidence of optimal transport benefits**
> In the included code with our submission, we have added a masking step before solving the optimal transport problem.
> Because each output pixel can only predict 2D locations within a limited spatial range, we mask out any candidate–target pairs where the target is outside the reachable range of the candidate pixel.
> This guarantees that, for any target, the optimal transport algorithm assigns only candidates that are in range of predicting the target’s location.
> We have clarified this component in the revised manuscript, line 913.
>
> We have conducted an additional experiment to assess the impact of choosing an appropriate maximal prediction range for candidates.
> We have trained several models with different maximal offset ranges.
> A range of 1 means that each output pixel can predict locations only on the exact surface it covers.
> In this regime, our masking procedure yields only a single candidate for each target; therefore, the optimal transport solution reduces to an identity pixel-wise mapping.
> Increasing the prediction range allows more output pixels to be valid candidates for each target, thereby making the optimal transport assignment relevant.
>
> We have reported results for prediction ranges from 1 to 3 pixels in a new figure, Figure 5, in Appendix 2.
> We have found best performance for a range of 1.5 times the output pixel size, equivalent to 3 times the pixel size of the original image.
> These findings support that optimal transport benefits our framework. It does not collapse to the identity mapping; otherwise all results would have been similar to the performance of the model with a predictive range of 1.

---

> > ### Author Response · Authors · 2025-11-21
> > **Response Part 2/3**
> >
> > **Non-Maximum Suppression.**
> > DECODE and LiteLoc both use a form of non-maximal suppression: indeed, DECODE describes adding an extra post‑processing step that uses two thresholds, which they refer to as, quoting, "a variant of non‑maximum suppression" (see "Methods $\rightarrow$ Obtaining localizations and post‑processing" in [1]).
> > Their variation introduces a second threshold that preserves high scoring pixels even when they are not local maxima.
> > LiteLoc's authors refer to a similar technique, though they do not call it a variant of NMS.
> >
> > This variant of NMS can be found in DECODE and LiteLoc repositories, see the following functions:
> >
> > - DECODE official implementation: https://github.com/TuragaLab/DECODE/blob/4c0f38c33681d1c3005bc18bd99a11f46d30f1c1/decode/neuralfitter/post_processing.py#L253
> >
> > - LiteLoc's reimplementation of DECODE : https://github.com/Li-Lab-SUSTech/LiteLoc/blob/ee00bad84f725aa7337984978ef06f59c1900651/network/decode.py#L233
> >
> > - Liteloc official implementation : https://github.com/Li-Lab-SUSTech/LiteLoc/blob/ee00bad84f725aa7337984978ef06f59c1900651/network/liteloc.py#L173
> >
> > The reviewer is correct that we should call this method an NMS‑*variant*, following the words used by DECODE's authors; we have updated our manuscript accordingly.
> >
> >
> > **End-to-end learning clarification.**
> > We refer to end‑to‑end learning as training all parts of the model together through a single objective closely related to the inference task.
> > This eliminates the need for heuristic, hand‑crafted post‑processing steps. This definition has been adopted by many object‑detection works before ours; see references [2,3].
> > In this context, our new loss function eliminates the variant of NMS used only at inference by DECODE and LiteLoc, which improves performance by bringing closer inference and training objectives, see Table 3.
> > After all, the impact of NMS on the final result is non negligible, since this process makes hard decisions by discarding detections based on a single fixed parameter. Discarding this layer allows to benefit from the flexibility of deep neural networks at the final processing step, which enhances the overall performance.
> >
> > **Connection between our contributions.**
> > Our two contributions are orthogonal: one is a loss-function improvement, the other is an deep neural network architecture improvement. Both can be used independently as shown in our ablation study Table 3, showing best performance when paired.
> >
> > **Training pipeline.**
> > We have clarified the nature of the objects manipulated by our network in the revised manuscript, Appendix 2.

---

> > > ### Author Response · Authors · 2025-11-21
> > > **Response part 3/3**
> > >
> > > **DeepLoco, Boyd et al., 2018.**
> > > We thank the reviewer for highlighting the work of Boyd et al. who also consider a set prediction formulation. Whereas we use
> > > optimal transport tools and a regularized Wasserstein loss function, they use a maximum mean discrepancy (MMD) criterion.
> > > Even though the loss and the corresponding algorithm are significantly different, this
> > > work is indeed highly relevant and we apologize for missing this reference.
> > > We have run the authors’ original code and obtained reasonable performance on low‑density setups (dataset MT0N1LDAS), yet those remain lower than DECODE's.
> > > Furthermore, at higher densities (dataset MT0N1HDAS), we could not achieve reasonable performance despite significant effort to tune both the simulation parameters and the sigmas of the MMD kernel during the brief period of the rebuttal.
> > > Here are our results:
> > >
> > > | Dataset   | Method   | Precision | Recall   | Jac      | RMSElat   | RMSEax    | E3D       |
> > > | --------- | -------- | --------- | -------- | -------- | --------- | --------- | --------- |
> > > | MT0N1LDAS | DeepLoco | 0.961(1)  | 0.975(8) | 0.941(6) | 29.2(2.3) | 26.3(1.7) | 0.755(20) |
> > > | MT0N1LDAS | DECODE   | 0.961(3)  | 0.998(1) | 0.959(3) | 8.8(01)   | 10.7(01)  | 0.895(3)  |
> > > | MT0N1HDAS | DeepLoco | 0.60(6)   | 0.67(7)  | 0.47(4)  | 130(5)    | 132(13)   | -0.14(7)  |
> > > | MT0N1HDAS | DECODE   | 0.923(3)  | 0.946(2) | 0.876(4) | 32.2(03)  | 33.0(04)  | 0.706(4)  |
> > >
> > >
> > > Furthermore, DeepLoco's authors admit that achieving reasonable results demands substantial effort to tune all simulation parameters. Quoting their `README`:
> > > > To try this on a new dataset you'll need [...] to make sure that the simulated data looks as similar as possible to the real data.
> > > This could be quite difficult: you'll need to adjust many hardcoded values in `empirical_sim.py`, as well as the generative model settings (in `train_script.py`). You might also need to modify the learning rate schedule of the network.
> > >
> > > While optimal transport is more popular for object detection these days than MMD, understanding why it performs better in the context of SMLM is an interesting theoretical question worth investigating in the future.
> > >
> > >
> > > **Minor comments.**
> > > We thank the reviewer for spotting typos.
> > > We have added a definition for $\text{RMSE}_\text{vol}$ line 372 and corrected citation issues in the revised manuscript.
> > > The mismatch between Table 1 and Table 3 is a mistake caused by rounding errors by hand. We have rerun the evaluation and have found a mean of 0.7501 with a standard deviation of 0.0038, so the correct rounding is 0.004 and not 0.003. We have updated Table 3 accordingly.
> > >
> > > [1] Speiser, A., Müller, LR., Hoess, P. et al. Deep learning enables fast and dense single-molecule localization with high accuracy. Nat Methods 18, 1082–1090 (2021).
> > >
> > > [2] Learning Non-Maximum Suppression, Jan Hosang, Rodrigo Benenson, Bernt Schiele; Proceedings of the IEEE Conference on Computer Vision and Pattern Recognition (CVPR), 2017, pp. 4507-4515
> > >
> > > [3] Q. Zhou and C. Yu, "Object Detection Made Simpler by Eliminating Heuristic NMS," in IEEE Transactions on Multimedia, vol. 25, pp. 9254-9262, 2023,

---

> > > > ### Comment · Reviewer_pzpv · 2025-11-28
> > > >
> > > > I appreciate the detailed rebuttal and the additional experiments. I understand that the proposed method’s effectiveness is empirically validated. However, I still feel that the paper does not yet articulate a sufficiently clear motivation or intuition for why and how the method works—specifically, how the OT loss and the refinement network address the fundamental ambiguities that arise under high-density conditions.
> > > >
> > > > At present, the paper introduces two technical ideas (an OT-based set loss and an iterative physics-aware refinement), but the conceptual link between:
> > > > * the decision problem intrinsic in SMLM,
> > > > * the limitations of prior NMS-like pipelines, and
> > > > * how OT + refinement jointly resolve these issues,
> > > > remains underdeveloped. This lack of conceptual clarity makes it harder for readers to fully understand the strength and significance of the contribution.
> > > >
> > > > Clarifying the following points would, in my view, significantly improve the clarity and impact of the paper:
> > > >
> > > >
> > > > 1. Why SMLM fundamentally requires a decision step, and why this step is reframed as set matching problem
> > > >     SMLM pipelines must convert a continuous detection-score map into a discrete set of emitters. Under high-density conditions, this requires simultaneously:
> > > >     (i) suppressing spurious local maxima, while
> > > >     (ii) not merging nearby emitters that fall within the same or adjacent pixels.
> > > >
> > > >     These objectives inherently conflict and are specific to SMLM (unlike standard object detection). Existing methods (e.g., DECODE, LiteLoc) resolve this using a two-threshold procedure (”variants of NMS”), which is heuristic and non-differentiable.
> > > >     However, the paper does not clearly explain that the decision step can be formulated as a set-matching problem between predicted candidates and true emitters. The current phrasing (”framing SMLM as a supervised-learning problem leads to a set matching formulation”) does not sufficiently convey this motivation.
> > > >
> > > >     If this reframing were made explicit, the choice of an OT-based loss would follow naturally, since OT directly handles matching between predicted and ground-truth sets.
> > > > 2. The motivation of the proposed method
> > > >     As noted in the rebuttal, a core motivation for using OT distance is to “bringing closer inference and training objectives” by incorporating the decision process—formulated as a set-matching problem—directly into the training loss. This is a compelling rationale, but it is not stated explicitly  in the paper.
> > > >
> > > >     If point 1. above is clarified, this motivation becomes far more convincing to the readers, even though the method still requires choosing a final detection threshold.
> > > >
> > > >     The additional motivation mentioned around line 76 regarding high-density case is also helpful and should be retained. Moreover, making more explicit use of the illustration in Appendix A.2—showing how multiple emitters within a single pixel may be resolved using predictions from surrounding pixels—could further strengthen the underlying intuition.
> > > > 3. The role of the refinement network
> > > >     The physics-based reconstruction step likely plays an essential role in localizing emitters from overlapping PSFs—particularly when combined with the OT loss (even if the contributions are technically orthogonal). Explicitly articulating this intuition would help readers understand why the proposed architecture outperforms prior methods, as well as the ablations reported in Table 3.
> > > >
> > > >     At present, the motivation is described mainly through an analogy to iterative refinement in optical flow, which does not convincingly justify its necessity in the context of SMLM. A clearer explanation of why this mechanism helps resolve high-density ambiguities—for example, that emitters too close to be separated based on local evidence alone can be localized by checking whether their predicted PSF superposition reproduces the input image (corresponding to problem 1-(ii)) —would significantly strengthen the paper.
> > > >
> > > > Overall, my impression is that the method has strong potential and performs well empirically. However, the paper would benefit greatly from presenting the underlying intuition and motivation more explicitly, ideally in the Introduction. Highlighting these conceptual points early on would improve clarity, strengthen the coherence of the contribution, and help readers better understand why the proposed formulation is effective.

---

> > > > > ### Author Response · Authors · 2025-12-02
> > > > >
> > > > > We thank the reviewer for their positive feedback on our rebuttal. We have uploaded a second revision of our paper with a modified introduction that should clarify the reviewer’s mentioned points. New edits are highlighted in orange.
> > > > >
> > > > > **SMLM decision step and NMS limitations.**
> > > > > Following the reviewer’s recommendations, we have updated our introduction to explicitly state that SMLM requires a decision step and that NMS‑based methods contain inherent flaws in this specific context:
> > > > >
> > > > > > Top methods [1,2] predict a detection map trained with pixel-wise objectives, and decide at inference whether a candidate exists or not by binarizing their map using a variant of non‑maximum suppression (NMS). This NMS-variant uses two thresholds to (i) suppress spurious local maxima while (ii) not merging nearby emitters. We see three main issues with this framework. (1) These pixel-wise loss functions do not account for multiple emitters within the same pixel. (2) Objectives (i) and (ii) are inherently in conflict, and this issue only worsens as density increases, where the probability of multiple emitters activating simultaneously at sub-pixel distance rises. (3) The precision–recall tradeoff is difficult to tune due to the two required hand-set thresholds. Figure~\ref{fig:motivations} in the Appendix illustrates problems (1) and (2).
> > > > >
> > > > > We have also added an extra figure, Figure 5, to illustrate the limitations of previous methods.
> > > > >
> > > > > **Motivations for using optimal transport.**
> > > > > We have modified our introduction to clearly express how our optimal transport loss function solves these issues, paired with our simpler decision method with a single threshold:
> > > > > > In this paper, we frame the SMLM training objective based on one-to-one matching between predicted and true emitters, using a new loss function constructed from optimal transport, and solve the decision problem at inference with a simple individual one threshold filtering. These changes solve problem (1) by removing pixel-wise assignments in the training objective, problem (2) by removing decision pipelines based on spatial proximity like NMS, and problem (3) by using a single threshold during filtering, which directly controls the precision-recall tradeoff. Furthermore, NMS non-differentiability prevents the model from optimizing for it: discarding it allows us to benefit from the flexibility of deep neural networks at the final model layer, unlocking end‑to‑end learning.
> > > > >
> > > > > **Motivations for the refinement network.**
> > > > > The motivations for using a refinement network are largely empirical. Those architectures have shown good results in previous work, mainly with optical flow estimation, but also with computer‑vision inverse problems, and simulation-based physics inverse problems; see our Related Work section.
> > > > > Following those successful works, we have created a differentiable simulator with respect to our decoder’s output and have built on the success of this paradigm to design our architecture, which in practice yields performance improvements.
> > > > >
> > > > > [1] Artur Speiser, Lucas-Raphael M¨uller, Philipp Hoess, Ulf Matti, Christopher J Obara, Wesley R
> > > > > Legant, Anna Kreshuk, Jakob H Macke, Jonas Ries, and Srinivas C Turaga. Deep learning enables
> > > > > fast and dense single-molecule localization with high accuracy. Nature Methods, 18:1082–1090,
> > > > > 2021
> > > > >
> > > > > [2] Yue Fei, Shuang Fu, Wei Shi, Ke Fang, Ruixiong Wang, Tianlun Zhang, and Yiming Li. Scalable and
> > > > > lightweight deep learning for efficient high accuracy single-molecule localization microscopy.
> > > > > Nature Communications, 16(7217), 2025

---

### Official Review · Reviewer_gn8m · 2025-11-02

**Soundness:** 4
**Presentation:** 3
**Contribution:** 3
**Rating:** 8
**Confidence:** 5

**Summary:**

Recent deep-learning methods for high-density single-molecule localization microscopy (SMLM) typically depend on non-differentiable post-processing steps, such as non-maximum suppression, to extract the final set of emitter coordinates. This paper reformulates emitter localization as a set-matching problem and introduces an entropy-regularized optimal transport loss that enables fully differentiable, end-to-end training without the need for heuristic post-processing. Furthermore, the proposed iterative refinement architecture integrates the microscope’s image formation model during inference, making the approach physics-aware and potentially enhancing robustness in real experimental conditions. Experiments on synthetic and real datasets demonstrate state-of-the-art performance, particularly in moderate and high emitter density regimes.

**Strengths:**

The paper is original in framing single-molecule localization as a set-matching problem and introducing an entropy-regularized optimal transport loss that removes the need for non-differentiable post-processing, enabling genuine end-to-end learning. The technical quality is high, with a carefully designed architecture that incorporates a physically accurate forward model of the imaging system, explicitly modeling shot, readout, and amplification noise. The results are clearly presented and demonstrate strong generalization from synthetic to real biological datasets, underscoring the method’s robustness and practical significance. Overall, the paper combines conceptual novelty, methodological rigor, and empirical clarity, marking a meaningful advance in the application of deep learning to super-resolution microscopy.

**Weaknesses:**

While the paper is strong overall, a few aspects could be clarified or expanded. First, the motivation for using an entropy-regularized optimal transport (OT) loss over standard bipartite matching approaches, such as the Hungarian assignment followed by pairwise regression losses (as employed in DETR and related works), remains insufficiently justified. A direct comparison or ablation highlighting the practical advantages of the entropic formulation—such as smoother gradients, faster convergence, or improved stability—would strengthen the claim. Second, the related work section omits several important prior efforts that jointly optimized the point-spread function (PSF) and the localization model for dense 3D SMLM, most notably DeepSTORM3D (Nehme et al., 2020). Including and discussing these works would provide a more balanced contextualization of the paper’s contributions. Finally, DECODE is excluded from the qualitative comparisons in Figure 4 under the justification that its results resemble LiteLoc; however, DECODE additionally provides per-emitter uncertainty estimates that can be leveraged for post-hoc filtering and might alter the visual comparison. A brief discussion or experiment acknowledging this aspect would help ensure a fairer and more complete evaluation.

**Questions:**

1) Could the authors clarify the motivation for adopting an entropy-regularized optimal transport loss (via the Sinkhorn–Knopp algorithm) instead of a DETR-style bipartite matching loss? In particular, what empirical or theoretical advantages (e.g., smoother optimization, differentiability, computational speed) motivated this choice, and how many Sinkhorn iterations were used in practice to approximate the transport plan?
2) How is the gradient of the Sinkhorn-based loss computed during backpropagation? Is there a closed-form analytical expression for the gradient, or is it obtained by unrolling the Sinkhorn iterations? Providing this clarification—ideally with a short note or derivation in the appendix—would significantly improve the clarity of the method.

---

> ### Author Response · Authors · 2025-11-21
> **Response**
>
> **Regularized optimal transport and bipartite matching.**
> We have adopted a formulation based on regularized optimal transport because it yields a differentiable, GPU‑friendly
> algorithm with fast convergence. We found that Sinkhorn's algorithm converges in fewer than 20 iterations, and its gradient is obtained automatically by sequential chain rules through the iterative updates.
> A practical implementation of Sinkhorn's algorithm can be found in the submitted code, in the file `src/smlm/losses/ot.py` at lines 61–76.
> Following the reviewer's remarks, we have added an extra appendix section (Appendix 4) in the revised manuscript evaluating performances, with the Hungarian algorithm addressing the standard bipartite matching formulation, and Sinkhorn's algorithm with different regularization parameters. It demonstrates that the latter is not only faster, but also achieves better performance, particularly for small regularization parameters.
>
> **Live optimization of the PSF.**
> As highlighted by the reviewer, DeepSTORM3D effectively optimizes its PSF in an end‑to‑end manner through its decoding‑encoding framework. Their objective, however, is fundamentally different from ours: they aim to design an “optimal Tetrapod PSF,” i.e., a PSF that facilitates the resolution of the SMLM inverse problem by minimizing emitter overlap and maximizing information about axial localization. Consequently, DeepSTORM3D acts as a co‑designer of the imaging system and addresses the problem from a microscope‑engineering perspective. In contrast, our work operates entirely a posteriori, after the acquisition process. We do not modify anything about the experimental setup; we are given an estimation of the PSF and the observed images as they were recorded.
>
> Within our framework, optimizing the PSF in the simulator would primarily serve to compensate for calibration mismatches between the available pre‑calibrated PSF and the real PSF of the images. This is relevant; please see our detailed response to Reviewer VfVn.
>
> **Uncertainty filtering.**
> The reviewer is correct that we do not provide an uncertainty estimate for each candidate.
> We have tried to replace the learned weights in our optimal transport loss with individual uncertainties predicted for each activation candidate, but this has caused the training to collapse.
>
> Filtering activations after inference by their estimated uncertainties adds an extra threshold to the inference process (or two if separate thresholds are used for axial and lateral uncertainties). This makes practical comparison in synthetic benchmarks difficult, because DECODE offers no default values for those thresholds, and good values may vary with image SNR.
> Consequently, we have omitted this filtering step while producing Table 1 and in the qualitative comparisons in Figure 4, in order to avoid any arbitrary choice that may bias results and to provide an objective comparison.
> The reviewer is correct that we have not emphasize this point sufficiently; we have added a discussion about this limitation in the revised manuscript, line 478.
>
> Finally, we have attempted to incorporate uncertainty estimation via localisation‑guided detection scores - following recent work such as [1] - but the results have been unsatisfactory. We agree with the reviewer that this line of research is valuable and will pursue it in future work.
>
> [1] Detection Transformer with Stable Matching; Shilong Liu, Tianhe Ren, Jiayu Chen, Zhaoyang Zeng, Hao Zhang, Feng Li, Hongyang Li, Jun Huang, Hang Su, Jun Zhu, Lei Zhang; Proceedings of the IEEE/CVF International Conference on Computer Vision (ICCV), 2023, pp. 6491-6500

---

### Official Review · Reviewer_4CME · 2025-11-02

**Soundness:** 3
**Presentation:** 3
**Contribution:** 3
**Rating:** 6
**Confidence:** 4

**Summary:**

The paper presents a methodology to train neural networks for detection of activations in single-molecule localization microscopy. The main premise of the paper is that optimal transport can serve as a tool to design loss functions that are fully differentiable and better performing for solving this problem. The proposed optimal transport objective matches activation points with detected observations in the images. The proposed architecture also incorporates an iterative refinement feedback which reconstructs the image with an image formation simulation given the model detections.

**Strengths:**

* Formulation of a differentiable loss function to detect activation points in single-molecule localization microscopy.
* Effective use of optimal transport theory to design the proposed method.
* Natural design of the iterative refinement strategy based on physical image-formation principles.
* Experimental evaluation with multiple performance metrics and relevant baselines.
* Qualitatively, the methodology also improves detections in real world data.

**Weaknesses:**

* While the application of optimal transport in SMLM is novel, it has been used before for similar detection and localization problems.
* The performance metrics seem to be saturated in the selected datasets for evaluation. The performance gains seem marginal.

**Questions:**

* What makes the formulation novel in the context of object detection or point localization?
* Does the SMLM problem really need a new methodology, given that other methods perform comparably?
* What other benefits (computational, speed, etc) can be obtained from the proposed formulation?
* Is this applicable to other microscopy imaging types?

---

> ### Author Response · Authors · 2025-11-21
> **Response**
>
> **Novel contributions.**
> While recent object‑detection and point‑localisation works have popularized the usage of optimal transport in machine learning, we believe that leveraging it for a real-world, physics‑driven application such as SMLM is non‑trivial. Key challenges we had to overcome include:
>
> 1. *Deriving an uncertainty‑aware multi‑quantity loss*. Solving SMLM requires to jointly predict five quantities per candidate activation ($x,y,z,n$, and detection score $s$) that involve different physical units and uncertainties (axial localisation is harder than lateral due to PSF shape, and photon count decreases the further a fluorophore is from the focal plane). Our loss dynamically adapts to these heterogeneous quantities, without one dimension adversely affecting the others.
> We are not aware of any other optimal transport loss function that handles multiple physical quantities and dynamically adapts to their respective uncertainties.
>
> 2. *Designing an efficient decoder.* Creating a decoder that delivers the sub‑pixel precision required by SMLM while remaining compatible with our optimal transport framework has proven non‑trivial. For instance, the vanilla transformer decoder from DETR yields poor results when applied to SMLM. Through experimentation, we found that incorporating CNN layers in the decoder, and blending optimal transport techniques with architectures similar to DECODE, was effective for this very specific application.
>
> 3. *Leveraging SMLM simulation inside the model architecture.* Training iterative network end‑to‑end with physics‑based simulation has been explored in other physical domains (see our Related Work section). However, to the best of our knowledge we are the first to leverage a fully differentiable SMLM simulator through an iterative refinement neural network and a optimal-transport loss function.
>
> **Research in SMLM.**
> We believe that even small advances in SMLM have great potential impact.
> Being able to increase tagged biomolecule density while maintaining precise localisation is the key bottle neck at the moment for the SMLM field. Dynamical evolution of organelles, and especially their geometry, has been overlooked and play a major role in biological function, for example in endoplasmic reticulum, see [3].
>
> Classical SMLM delivers super‑resolved 3D images at nanometer resolution, but as stated in our introduction, its usefulness for biology is limited by the speed at which data can be collected. Even a modest increase in fluorophore density yields information that was previously inaccessible and improves live‑cell acquisition. Because protein function depends on both chemical composition and 3D configuration, improving 3D accuracy may translates directly into new biological insight.
> Thus, refining SMLM to tolerate higher densities remains a worthwhile research direction [1,2], even if improvements are not major.
>
> **Benefits of our formulation.** Our formulation offers two main benefits:
>
> 1. The optimal transport loss function lets us work with set‑based predictions natively during training, eliminating the need for a two-stage procedure as in DECODE and LiteLoc; these methods rely on auxiliary pixel‑wise maps during training and variants of NMS layers at inference to convert maps into sets.
>
> 2. In deep learning models supervised by simulated images like ours, prior knowledge about the SMLM problem comes from the simulator that is informed by the physics associated to optics. Our iterative architecture leverages the simulator’s design directly inside the network, providing an efficient way to incorporate problem‑specific information and enhanced robustness to the ambiguous conditions we seek to solve, i.e. high density of tagged biomolecules.
>
> Together with superior benchmark performance, we believe these advantages make our approach both simpler and more effective for SMLM than previous deep learning approaches.
>
> **Impact on other microscopy techniques.**
> We have designed our method specifically for SMLM, where detection and localisation are performed on sequences of 2D images. It could also be useful for other imaging frameworks that identify and localize individual point emitters from camera‑based data, but doing so would require significant work to build a differentiable simulator for those domains. As we are not trained in those fields, we prefer not to make strong claims about the transferability of our work to other microscopy modalities or unrelated domains.
>
> [1] Lelek, M., Gyparaki, M.T., Beliu, G. et al. Single-molecule localization microscopy. Nat Rev Methods Primers 1, 39 (2021)
>
> [2] Cabillic, M., Forriere, H., Bettarel, L. et al. In-depth single molecule localization microscopy using adaptive optics and single objective light-sheet microscopy. Nat Commun 16, 8362 (2025)
>
> [3] Obara, C.J., Nixon-Abell, J., Moore, A.S. et al. Motion of VAPB molecules reveals ER–mitochondria contact site subdomains. Nature 626, 169–176 (2024)

---

### Official Review · Reviewer_VfVn · 2025-11-04

**Soundness:** 3
**Presentation:** 4
**Contribution:** 3
**Rating:** 6
**Confidence:** 4

**Summary:**

The paper treats the problem of single molecule light microscopy reconstruction as a set matching problem, which is a sound strategy. The authors develop an optimal transport loss and a model architecture to solve this problem. The strategy and results are promising, but the robustness and generalization of the method need to be evaluated further.

**Strengths:**

This work introduces two key computational innovations to single-molecule localization microscopy (SMLM). First, the authors reformulate SMLM as a set-matching problem and derive an optimal transport loss function, enabling end-to-end differentiable training . This is the first application of optimal transport theory to SMLM, to the best of my knowledge, and represents a principled approach to handling variable-size sets of fluorophore detections. Second, they propose an iterative refinement architecture that explicitly integrates knowledge of the microscope's optical system by simulating expected frames from current predictions using the known point spread function (PSF), allowing computation of loss between experimental data and estimated molecular distribution.  The combined approach achieves state-of-the-art performance on the E3D metric across multiple density regimes, with particularly strong improvements in localization accuracy (lowest RMSE) at high densities (density 8.0), demonstrating the method's potential to enable faster acquisition times for live-cell imaging.

**Weaknesses:**

The following limitations can constrain the practical applicability of current method. First, the method exhibits consistently lower recall than competitors across all experiments (Table 1), meaning it misses more true fluorophores despite having excellent precision—this trade-off could lead to incomplete biological reconstructions. It is not clear from the paper which aspects of data or optimization influence precision and recall.

Second, the approach requires precise PSF calibration using fluorescent beads and the method's robustness to PSF miscalibration or drift remains unexplored. The iterative architecture incurs 3× higher computational cost during training (20 hours) and slower inference compared to single-pass models.

Third, the high-density validation on real biological data uses temporal binning as a proxy, which artificially improves signal-to-noise ratio through averaging rather than testing true high-density, low-SNR conditions that would occur in practice. Majority of training relies on synthetic simulations with uniform priors (10-30 activations, uniform spatial distribution), which may not capture the structured, heterogeneous distributions in real biological specimens.

**Questions:**

Majority of my concerns are centered on evaluation of the robustness of the method to unwanted aberrations in imaging, density of fluorophores, and quantum efficiency of fluorophores.

I suggest a systematic series of simulations and experiments to address these. For example,
1. Consider adding a learned layer to the image formation model that learns a single 3D kernel that allows the image formation model to adapt aberrations in the data.
2. Simulate data with varying quantum efficiencies and aberrations different from training data, and evaluate the sensitivity of a trained model to these imperfections in real data.
3. Evaluate feature activations at the end of the encoder and at the end of the decoder for specific raw data frames to test that the estimated localizations make sense, and models are truly learning a relationship between the intensity distributions and the localization of emitters.

---

> ### Author Response · Authors · 2025-11-21
> **Response part 1/2**
>
> **Precision-recall trade-off.**
> At inference time, we filter candidate activations with a single threshold, allowing to control the precision‑recall trade‑off. Following the reviewer’s remark, we have clarified this in the main text, line 349. In contrast, DECODE and LiteLoc use a variant of an NMS layer controlled by two thresholds, which is harder to tune.
>
> **Learning PSF during training.**
> We agree that adapting the PSF during training is an important direction.
> However, full PSF characterisation is extremely challenging as part of its properties are linked to optical process associated to the cells. We are pursuing a direction in which the design of the microscope will be optimized to ensure robust PSF structures and stable deconvolution of images, but we believe it falls outside the scope of the present paper.
>
> **Computational cost**
> We reported the computational footprint of our method in Appendix 3. Our method runs roughly three times slower than DECODE and LiteLoc. Nevertheless, it outperforms both baselines in terms of $E_{\text{3D}}$ after only $2\%$ of the total training ($\simeq$ 2000 steps, or about 20min on a 6000 ADA GPU). The backward pass through successive iterations increases VRAM requirements: a batch size of 128 3x64x64 images takes 80GB vs 15GB for DECODE, but we believe this should not deter adoption, given that modern GPUs are getting increasingly accessible and do not represent a significant investment for laboratories relative to the cost of usual microscopy equipment.
>
> **Robustness to PSF and simulation parameters mismatches.**
> Concerning PSF mismatch, note that the ground‑truth PSF used to synthesize the dataset is not public; like other authors, we had to estimate it from noisy acquisitions of synthetic beads.
> Therefore, all results in Table 1 inherently contain a PSF mismatch.
> Besides, full characterisation of the PSF is not possible, due to the cells' optical properties that is activity and cellular function dependant and due to non-characterizable optical imperfections in real microscope.
>
> Regarding simulation parameters, we already apply random multiplicative jitter to the camera parameters during training to achieve better robustness: each parameter is scaled by $e^{\rho}$ with $\rho\sim\mathcal N(0,0.03)$. This data augmentation step was not mentioned in the original submission but is present in the included code; we have clarified it in line 333.
>
> Following the reviewer’s question about the robustness of our model to imperfect simulation parameters, we have reported performance in a new appendix section, Appendix 5, for varying camera parameters, including quantum efficiency, electromagnetic gain, and all parameters listed in Table 4. Our method demonstrates good resilience to these deviations.
>
> **Increased SNR at very high density.**
> We thank the reviewer for highlighting this issue, which we had already noted in our original submission.
> To address it, we have redesigned our 8.0-dataset: we have added extra noise to each newly binned frame using the camera model. The added noise, together with the residual noise that remains after binning, guarantees that the resulting image contains more noise than the original, yielding a lower SNR than would be expected from a raw acquisition. We have added a paragraph in the revised paper at line 426 to clarify this procedure, and have updated Table 1 accordingly; our primary conclusions remain unchanged.

---

> ### Author Response · Authors · 2025-11-21
> **Response part 2/2**
>
> **Unrealistic uniformly-distributed coordinates prior.**
> We share the reviewer’s view that specific priors would be valuable when addressing specific biological problems. Globally, it is a tradeoff between generality of the method, stability of the organelles structures, and targeted experiences.
>
> Obtaining priors could be achieved only on structures persenting both geometrical stability and dynamical tagging stability which is rare in the cell that is a non-equilibirum dynamical system.
>
> For example, the authors of [1] have illustrated the potential negative impact of a specialised prior by simulating training images with fluorophores located on artificial filaments to induce a structural prior for microtubules. While their microtubule reconstruction is excellent, the method suffers from hallucinations when applied to other structures such as nuclear pore complexes (see their Supplementary Figure 13).
>
> Hence, we have opted for a uniform prior distribution, to avoid favoring any particular structure and removes the need to engineer dedicated priors for every structure. These concerns are shared with the authors of DECODE and LiteLoc, who also used a uniform prior. We are nevertheless currently investigating the use of biologically-inspired priors for future work.
>
> **Exploration of the input-output relation.**
> We are not sure to understand what experiment the reviewer recommends.
> We will be happy to perform additional experiments after clarification.
>
> [1] Ouyang, W., Aristov, A., Lelek, M. et al. Deep learning massively accelerates super-resolution localization microscopy. Nat Biotechnol 36, 460–468 (2018).

---

### Author Response · Authors · 2025-11-21
**Revision Summary**

We thank all reviewers for their clear and detailed reviews.
We have addressed each of their comments in individual responses and have uploaded a revised version of the manuscript, which includes additional experiments, clarifications, and corrected typos.
All major modifications are highlighted in blue, making them easy for the reviewers to identify.

Concise summary of the main modifications in the revised manuscript:
- We have improved our introduction and related work by explicitly stating how pixel-wise objectives can be detrimental to the final prediction and by motivating the use of optimal transport.
- We have added an extra appendix, Appendix 4, providing details about the implementation of the regularized optimal transport. We have run additional experiments, reported in Table 7, to compare the Hungarian algorithm with Sinkhorn's algorithm and to evaluate the impact of various regularization parameters.
- We have clarified our data augmentation process and added a new appendix, Appendix 5, to assess robustness to mis-configured parameters. We have demonstrated the robustness of our method with additional experiments reported in Table 8.
- We have added technical details about our network architectures in the main text, clarified our decoder architecture design in Appendix 2, and motivated the choice of the number of candidates $d$ with additional experiments reported in Table 5.
- We have clarified the parameters influencing the detection-localization tradeoff in Section 4. We have detailed how it can be effectively tuned with a single parameter by the end user, and we have clarified our automatic threshold selection strategy.
- We have added an extra experiment in Figure 5, reporting performances for variable values of the maximal offset range magnitude, which demonstrate the effectiveness of our optimal transport approach compared to the pixel-wise approach.
- We have revisited our temporal binning for the dataset with a density of 8.0 in Table 1 to prevent SNR improvement, and have updated the results accordingly.
- We have improved our conclusion.
- We have fixed several typos and improved some phrasing throughout the manuscript.

We would be glad to clarify any remaining points, and appreciate any additional feedback from the reviewers.

---

### Meta-Review · Area_Chair_QPWJ · 2025-12-09

**Summary:**

Dear authors,

The reviewers were, mostly positive about this submission. After reading the reviews, your responses, and the paper itself, I am following their recommendation. The paper has a very clear focus (how to improve source-localisations for single-molecule localisation microscopy) and provides a clear answer to it by formulating it as a set-matching problem, and provides empirical evidence of its utility. The paper addresses an important problem, and making the extraction step differentiable is both elegant and useful. Reviewers (mostly) agreed that the technical quality is high.

Please do take the input of the reviewers into account when preparing your final submission. In particular (but not only), provide clear discussions of the limitations of the methods in the final version (e.g. from VfVn), and also discussion of related work (gn8m, and relationship with DeepLoco). One reviewer was critical, but I do not think that it provides sufficient evidence for not accepting the paper-- they do make useful suggestions on how to improve the clarity of the paper.

I hope you find this feedback useful, and congratulations on this paper!

**Reviewer Concerns:**

see above.

**Reviewer Scores:**

I am unwilling to follow this request. The ACs are being asked to perform an unreasonable amount of extra work this year. In particular, I understand the concern about potential collusion. However, I do think that asking ACs to now read a whole new set of papers, reviews and (extremely long...) rebuttals and to try to condense them into decisions without a chance for discussions is both asking a lot from us, and will, inevitably, result in (on average) poorer decisions for everyone.

I do not see any value in spending even more time to try to 'guess' how each reviewer would have changed their mind or not. I am just trying to make appropriate decisions on the paper and focus on the science. The authors can see the reviewer inputs and should take it into account.

---

### Decision · Program_Chairs · 2026-01-26

Accept (Poster)